# MATEXPERT: DECOMPOSING MATERIALS DISCOVERY BY MIMICKING HUMAN EXPERTS

**Qianggang Ding**[1,2]**, Santiago Miret**[3]**, Bang Liu**[1,2]*

[1] DIRO & Institut Courtois, Université de Montréal
[2] Mila - Quebec AI Institute [3] Intel Labs
{qianggang.ding, bang.liu}@umontreal.ca
santiago.miret@intel.com

## ABSTRACT

Material discovery is a critical research area with profound implications for various industries. In this work, we introduce *MatExpert*, a novel framework that leverages Large Language Models (LLMs) and contrastive learning to accelerate the discovery and design of new solid-state materials. Inspired by the workflow of human materials design experts, our approach integrates three key stages: retrieval, transition, and generation. First, in the retrieval stage, MatExpert identifies an existing material that closely matches the desired criteria. Second, in the transition stage, MatExpert outlines the necessary modifications to transform this material formulation to meet specific requirements outlined by the initial user query. Third, in the generation state, MatExpert performs detailed computations and structural generation to create new materials based on the provided information. Our experimental results demonstrate that MatExpert outperforms state-of-the-art methods in material generation tasks, achieving superior performance across various metrics including validity, distribution, and stability. As such, MatExpert represents a meaningful advancement in computational material discovery using langauge-based generative models. Codes are available at: https://github.com/BangLab-UdeM-Mila/MatExpert

## 1 INTRODUCTION

The discovery and design of new materials are central challenges in modern materials science, driven by the need for materials with tailored properties for applications in energy, electronics, and catalysis. Traditional methods for material discovery, such as high-throughput experiments and density functional theory (DFT) simulations, are computationally expensive and often require significant domain expertise to achieve accurate predictions (Miret et al., 2024). Recent advancements in artificial intelligence (AI), particularly large language models (LLMs), have opened new possibilities for automating and accelerating the materials design process (Miret & Krishnan, 2024; Jablonka et al., 2024; Song et al., 2023a;b; Zhang et al., 2024; Ramos et al., 2024).

LLMs such as GPT-4 OpenAI (2023) have demonstrated remarkable success in natural language processing tasks and have shown potential for application in scientific problems beyond language, including chemistry and materials science Flam-Shepherd & Aspuru-Guzik (2023); Gruver et al. (2024); Schilling-Wilhelmi et al. (2024); Mirza et al. (2024); Delétang et al. (2023). For example, LLMs have been used to generate molecular structures Gruver et al. (2024) and predict material properties from textual descriptions Alampara et al. (2024). In the context of material generation, models like Crystal Diffusion Variational Autoencoders (CDVAE) Xie et al. (2022b), as well as fine-tuned LLMs such as LLaMA-2 Gruver et al. (2024), CrystaLLM Antunes et al. (2023), and LM-AC/LM-CH, have made significant progress in generating crystal structures directly from Crystallographic Information Files (CIFs) Antunes et al. (2023). Together, these models aim to accelerate materials design by predicting stable structures and optimizing key properties, such as energy above hull for stability Riebesell et al. (2023).

---

*Corresponding Author. Canada CIFAR AI Chair.

Despite the significant progress made in applying AI models to materials science, current methods for material generation still face critical limitations Miret & Krishnan (2024); Alampara et al. (2024); Mirza et al. (2024). Most approaches generate material structures in a single step, often resulting in static outputs that lack the flexibility for iterative refinement and optimization based on intermediate feedback. Moreover, these models struggle to integrate multimodal data—such as textual descriptions of desired properties and structural information like atomic coordinates from Crystallographic Information Files (CIFs)—in a meaningful and cohesive way. Consequently, they fall short of replicating the complex, reasoning-driven processes employed by human experts, limiting their ability to generate materials that meet specific target properties.

In contrast to AI models, human experts follow a methodical, multi-step process when designing new materials. They begin by identifying an existing material with properties closely matching the desired specifications. From there, they apply a series of logical modifications—such as altering atomic arrangements or chemical compositions—to iteratively refine the material. Only after extensive refinement do they finalize the structural details, ensuring that the material is both feasible and aligned with the target properties.

To overcome the limitations of current models, we propose *MatExpert*, a framework that mimics this expert-driven workflow by breaking down the material generation process into three stages: retrieval, transition, and generation. Using a text-structure retrieval approach combined with chain-of-thought reasoning Wei et al. (2022), *MatExpert* can iteratively refine materials based on intermediate feedback, leading to more accurate and interpretable material designs that meet desired specifications.

To comprehensively assess the performance of *MatExpert*, we assembled a large-scale dataset from the NOMAD database Scheidgen et al. (2023), which contains a total of $2,886,120$ materials. This extensive evaluation on a large-scale testbed demonstrates the robustness and generalizability of *MatExpert* in addressing real-world materials discovery challenges.

Concretely, our paper makes the following contributions:

1. We propose *MatExpert*, a novel framework that mimics the expert-driven workflow for material discovery by decomposing the generation process into three stages: retrieval, transition, and generation. By leveraging a text-structure retrieval mechanism and a chain-of-thought reasoning approach, *MatExpert* enables the design of accurate and interpretable materials.

2. We curated a large-scale dataset from the NOMAD Scheidgen et al. (2023) database, which serves as a comprehensive testbed for evaluating *MatExpert* across diverse material compositions. We will publicly release the datasets and source codes upon publication.

3. We conducted extensive experiments on both the Material Project Jain et al. (2013) and the curated NOMAD datasets. Our results demonstrate that *MatExpert* significantly outperforms state-of-the-art baseline methods in material generation, showcasing its effectiveness and generalizability.

## 2 RELATED WORK

The use of AI and large language models (LLMs) in materials science has gained traction as a means to accelerate material discovery, which traditionally relies on computationally intensive methods like DFT simulations Duval et al. (2023); Govindarajan et al. (2024). Miret & Krishnan (2024) provide an extensive review of the limitations of current LLMs in materials science, particularly their inability to handle multi-modal data and iteratively refine structures based on feedback.

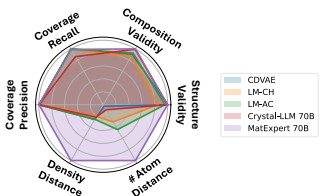

Early work, such as the Crystal Diffusion Variational Autoencoder (CDVAE) Xie et al. (2022b), generated crystal structures by incorporating physical stability constraints to produce valid periodic materials. Recent work has expanded on CDVAE's approach by exploring alternative representations for diffusion-based models Jiao et al. (2024); Yang et al. (2024); Zeni et al. (2023), as well as flow-

Figure 1: MatExpert achieves remarkable performance on all metrics, especially metrics of distances. See Table 1 for details.

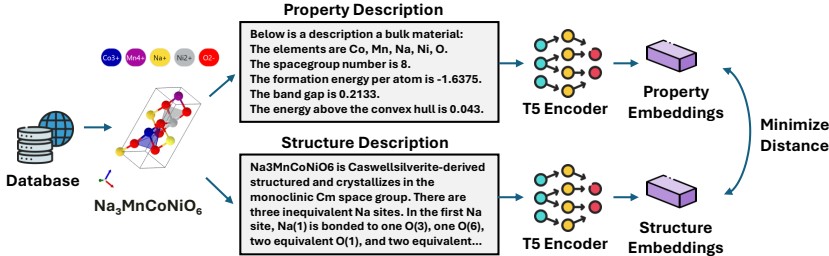

Figure 2: We utilize a contrastive learning framework to train two encoders for reference material retrieval. For a given sample material (e.g., Na₃MnCoNiO₆), we extract both its property description and structural description using PyMatgen Ong et al. (2013) and Robocrystallographer Ganose & Jain (2019), respectively. The model employs two T5-based encoders Raffel et al. (2020), which are trained to minimize the distance between these two representations.

matching models Miller et al. (2024). Further advancements include the fine-tuning of LLMs for materials generation. Gruver et al. (2024) fine-tuned LLaMA-2, improving metastable material generation rates, while CrystaLLM Antunes et al. (2023) employs an autoregressive model trained on CIF files to generate inorganic crystal structures, leveraging Monte Carlo Tree Search for refinement. Flam-Shepherd & Aspuru-Guzik (2023) proposed LM-AC and LM-CH, which explore generating 3D molecular structures directly from CIF, XYZ, and PDB formats, pushing the boundaries of LLM applications in material science. In addition to generative models for materials discovery, as well as the development domain-specific LLMs for diverse language tasks materials science (Song et al., 2023b;a; Xie et al., 2023; Zhang et al., 2024), LLMs are also being used to automate the extraction of structured data from scientific literature, streamlining the creation of datasets for machine learning Hira et al. (2024); Mishra et al. (2024); Schilling-Wilhelmi et al. (2024).

In summary, while LLMs show great promise, challenges remain in integrating multi-modal data and refining material designs iteratively. Most prior work has focused on unconditioned material generation, but current methods lack the ability to iteratively refine and reason like human experts. Our *MatExpert* framework addresses this shortcoming by mimicking an expert-driven workflow, employing multi-stage iterative refinement through retrieval, transition, and generation processes, and achieves remarkable performance compared to state-of-the-art methods, particularly in metrics of distances (see Figure 1).

## 3 METHODOLOGY

Our *MatExpert* framework simplifies the complex process of material discovery by emulating human experts through three key stages: **retrieval**, **transition**, and **generation**. First, we retrieve a reference material from a database that closely aligns with the desired properties. Next, using a fine-tuned large language model (LLM) with Low-Rank Adaptation (LoRA) Hu et al. (2022), we generate insights on how to modify the reference material to meet the target specifications. Finally, we generate the new material's structure based on the reference and transition steps. The framework employs a T5-based Raffel et al. (2020) model for the retrieval stage and a fine-tuned LLM for the transition and generation stages. A comprehensive illustration of the MatExpert framework pipeline is shown in Figure 3.

### 3.1 FIRST STAGE: RETRIEVAL

The retrieval stage is the foundational step of our framework, aimed at identifying the material in the database that best matches the desired property description provided by the user.

To bridge the gap between the representation of the desired properties and the relevant material, we apply contrastive learning. Contrastive learning Chen et al. (2020) is a self-supervised method that learns representations by bringing positive pairs closer together while pushing negative pairs farther apart in the embedding space. Contrastive learning was first applied in the domain Radford et al. (2021); Singh et al. (2022), recent works have successfully applied similar methods to domains to

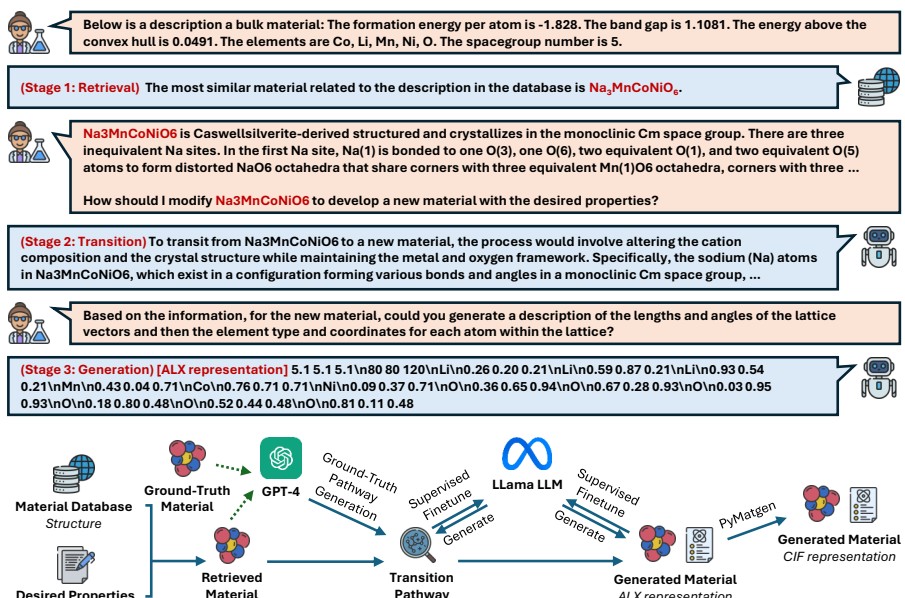

Figure 3: Pipeline of MatExpert: Given a description of the desired material, MatExpert first retrieves the most similar material from the database (e.g., $Na_3MnCoNiO_6$). Next, the LLM provides transition pathways to modify the retrieved material into the desired material (e.g., replacing Na with Li). Finally, the LLM generates the detailed structural information of the desired material ($Li_3MnCoNiO_6$). See Figure 7 for a full case of conditional material generation.

materials design, such as molecular Liu et al. (2022) and protein modeling Xu et al. (2023). In the context of material retrieval, we use this method to embed material properties and structures in a shared space, enabling the efficient retrieval of materials that closely match specific queries.

To implement this, we employ a contrastive learning framework (see Figure 2), using two parallel T5-based encoders to process and embed both property and structure descriptions. The property descriptions include key characteristics such as composition, space group number, formation energy, band gap, and energy above hull. Structural descriptions are extracted using RoboCrystallographer, which provides detailed linguistic descriptions of the structure for each material.

The primary objective is to ensure that the embedding of the property description ($\mathbf{q}_i$) is closely aligned with the embedding of the corresponding structure description ($\mathbf{m}_i$), while distinctly separating it from embeddings of other materials ($\mathbf{m}_k$). To quantitatively enforce this alignment, a contrastive loss function is defined as:

$$\mathcal{L} = -\log \frac{\exp(\text{sim}(\mathbf{q}_i, \mathbf{m}_i)/\tau)}{\sum_{k=1}^{N} \exp(\text{sim}(\mathbf{q}_i, \mathbf{m}_k)/\tau)}. \tag{1}$$

Here, $\text{sim}(\mathbf{q}_i, \mathbf{m}_i)$ denotes the cosine similarity between the property and structure embeddings, $\tau$ is a temperature parameter that controls the sharpness of the distribution, and $N$ is the size of negative sampling. The output of this stage is the best matching material, which serves as a candidate for further refinement.

## 3.2 SECOND STAGE: TRANSITION

The transition stage focuses on developing a detailed and viable method for modifying the retrieved material to meet the desired properties. This stage bridges the gap between the existing material and the target material by outlining specific structural or compositional changes required to achieve the desired specifications.

To build a model capable of generating transition pathways that describe how to transform the existing reference material into the desired target material, two key steps are necessary: (1) constructing a training dataset consisting of `<source material, transition pathways, target`

material> triples, and (2) fine-tuning a model that can produce `transition pathways` given an `source material` as a reference.

In this stage, we utilize two LLMs in a sequential process. First, during the training phase, we employ GPT-4 to generate ground-truth transition pathways, which provide comprehensive, step-by-step instructions for transforming the retrieved material to meet the target properties. Subsequently, we fine-tune a second, smaller LLM using these ground-truth pathways. In practical applications, including our experiments, this fine-tuned LLM is deployed to generate the potential transition pathways, offering a more efficient solution while maintaining accuracy.

**Ground-Truth Pathway Generation with GPT-4:** During the training phase, we utilize GPT-4 to generate detailed transition pathways by providing it with carefully designed prompts. These prompts are constructed to ensure that GPT-4 receives all necessary information about both the source material and the target material, enabling it to produce accurate and comprehensive modification pathways. An example template of such a prompt is:

```
I have two materials:  <formula_source> and <formula_target>.
Based on the descriptions and properties of the two materials
below, can you summarize the main reasons for the differences
in properties when transitioning from <formula_source> to
<formula_target>?  The description of <formula_source>:
<description_source>.  The description of <formula_target>:
<description_target>.
```

**Fine-Tuning the Second LLM:** The ground-truth transition pathways generated by GPT-4 form the basis for the supervised fine-tuning of a second LLM. This model is trained to learn and replicate the detailed modification steps provided by GPT-4, allowing it to generate potential transition pathways in practical applications. An example prompt template used for fine-tuning is:

```
I am looking to design a new material with the following
properties:  <property_list>.  The closest existing material
I found in the database is <formula_source>, which has
similar properties.  Below is the structure description of
<formula_source>.  \n <description_source> \n How should I modify
<formula_source> to develop a new material with the desired
properties?
```

## 3.3 THIRD STAGE: GENERATION

The generation stage is focused on producing the CIF (Crystallographic Information File) representation of the predicted material. Specifically, we first generate a detailed description of the lattice vectors and atomic coordinates, which we refer to as the ALX representation. This ALX representation is then automatically converted into CIF format using the `pymatgen` library, which serves as the final output of our framework.

**Fine-Tuning:** The generation process builds upon the conversation initiated during the earlier stages of the framework. The output from the transition stage serves as input for the generation stage. An additional prompt is introduced to extend the conversation, guiding the model to generate the ALX representation based on the transition pathway provided in the previous stage. The prompt for the generation stage is:

```
Based on the information, for the new material, could you generate
a description of the lengths and angles of the lattice vectors
and then the element type and coordinates for each atom within the
lattice?
```

**ALX Representation:** After fine-tuning, the LLM generates the ALX representation for the material based on the transition pathways (as shown in the final response in Figure 3). This representation includes:

- **Lattice Vectors:** The lengths and angles of the lattice vectors that define the unit cell of the material.

- **Atomic Coordinates:** The precise atomic types and their coordinates within the lattice, ensuring that the generated structure aligns with the desired material properties.

**Final Output in CIF Format:** Once the ALX representation is generated, it is converted into CIF format using the `pymatgen` library. The CIF file contains the complete structural information of the predicted material and serves as the final output of the generation stage. This file can be utilized for further computational studies, experimental validation, or integration into material databases.

## 4   DATA COLLECTION & PRE-PROCESSING

To rigorously evaluate the performance of *MatExpert*, we assembled a comprehensive dataset from the NOMAD database Scheidgen et al. (2023), utilizing its API. This extensive dataset serves as the large-scale testbed for assessing the capabilities of our framework.

**Data Collection from NOMAD**   NOMAD Scheidgen et al. (2023) is a vast repository of material data. For our study, we refined the dataset by excluding materials with inert elements to focus on chemically active compounds. We also restricted entries to those simulated using VASP Kresse & Hafner (1993) for high-quality computational data and excluded structures with over 30 atoms to enhance computational efficiency.

By applying these filtering criteria, we obtained a dataset of $2,886,120$ materials. This large-scale dataset provides a rich foundation for training and evaluating our models, enabling us to explore a wide range of material compositions and properties.

**Pre-processing**   Each filtered material was converted into CIF format, ensuring the structural integrity of the original data was preserved. CIF files are widely used in materials science to represent crystal structures and are compatible with various computational tools, facilitating seamless integration into our framework.

In addition to structural data, we enriched our dataset with linguistic descriptions of each material, generated using RoboCrystallographer Ganose & Jain (2019), a tool that automatically produces human-readable summaries of crystallographic information. This step allows for the integration of language-based processing within our framework, leveraging the strengths of large language models. We also extracted key material properties—such as formation energy, total energy, and elemental composition—using the M3GNet Chen & Ong (2022) and PyMatgen Ong et al. (2013) libraries in Python. These properties provide essential inputs for condition-based material design and evaluation within *MatExpert*.

By leveraging this meticulously curated dataset, we ensure that *MatExpert* is evaluated on a robust and diverse set of large-scale materials, facilitating comprehensive assessments of its performance and effectiveness.

## 5   EXPERIMENTS

### 5.1   DATASETS AND BASELINES

**Material Project:** We leverage datasets from the Materials Project database similar to prior studies Xie et al. (2022b); Gruver et al. (2024). The MP-20 dataset (Xie et al., 2022b) for unconditional generation contains 45,231 stable materials, filtered to exclude structures with over 30 atoms per unit cell for computational efficiency. The datasets are exclusively used for the unconditional generation task.

**NOMAD:** We also collected a large-scale dataset from the NOMAD database, consisting of 2,928,355 materials. The dataset was filtered to exclude structures with inert atoms and more than 30 atoms per unit cell, and it only includes structures computed using VASP in the database. The datasets are exclusively used for the conditional generation task.

To evaluate the effectiveness of MatExpert, we compare its performance against the following state-of-the-art baselines in material generation.

Table 1: MatExpert shows general outperformance compared to baseline methods for unconditional generation on Materials Project based on various metrics like validity checks, coverage, property distribution, and stability. The best results are in **bold**, while the second-best results are underlined.

| Method | Validity Check | | Coverage | | Distribution | | Metastable | Stable |
|---|---|---|---|---|---|---|---|---|
| | Struc↑ | Comp↑ | Rec↑ | Prec↑ | wdist($\rho$)↓ | wdist($N_{el}$)↓ | M3GNet↑ | DFT↑ |
| CDVAE | **1.00** | 0.867 | 0.991 | **0.995** | 0.68 | 1.43 | 28.8% | 5.4% |
| LM-CH | 0.848 | 0.835 | 0.992 | 0.978 | 0.86 | 0.13 | n/a | n/a |
| LM-AC | 0.958 | 0.889 | **0.996** | 0.985 | 0.69 | 0.09 | n/a | n/a |
| **Crystal-LLM with LLaMA-2** | | | | | | | | |
| 7B ($\tau = 1.0$) | 0.918 | 0.879 | 0.969 | 0.960 | 3.85 | 0.96 | 35.1% | 6.7% |
| 7B ($\tau = 0.7$) | 0.964 | 0.933 | 0.911 | 0.949 | 3.61 | 1.06 | 35.0% | 6.2% |
| 70B ($\tau = 1.0$) | 0.965 | 0.863 | 0.968 | 0.983 | 1.72 | 0.55 | 35.4% | 10.0% |
| 70B ($\tau = 0.7$) | 0.996 | 0.954 | 0.858 | 0.989 | 0.81 | 0.44 | 49.8% | 10.6% |
| **MatExpert with LLaMA-2** | | | | | | | | |
| 7B ($\tau = 1.0$) | 0.920 | 0.908 | 0.984 | 0.994 | 0.49 | 0.11 | 37.5% | 7.8% |
| 7B ($\tau = 0.7$) | 0.946 | 0.943 | 0.952 | 0.986 | 0.51 | 0.13 | 36.2% | 8.0% |
| 70B ($\tau = 1.0$) | 0.968 | 0.878 | 0.982 | 0.986 | 0.23 | 0.06 | 50.2% | 11.3% |
| 70B ($\tau = 0.7$) | 0.998 | 0.959 | 0.969 | 0.993 | 0.23 | 0.07 | 50.9% | 11.8% |
| **MatExpert with LLaMA-3** | | | | | | | | |
| 8B ($\tau = 1.0$) | 0.933 | 0.910 | 0.992 | **0.995** | 0.36 | 0.07 | 39.1% | 8.1% |
| 8B ($\tau = 0.7$) | 0.975 | 0.959 | 0.988 | 0.990 | 0.38 | 0.10 | 39.5% | 8.4% |
| 70B ($\tau = 1.0$) | 0.970 | 0.880 | 0.980 | 0.988 | 0.21 | 0.05 | 50.5% | 11.0% |
| 70B ($\tau = 0.7$) | 0.998 | **0.961** | 0.986 | 0.991 | **0.18** | **0.04** | **51.0%** | **12.0%** |

Table 2: Comparison results of conditional generation on NOMAD datasets with MatExpert outperforming baseline methods. The best results are in **bold**.

| Method | Validity Check | | Coverage | | Distribution | |
|---|---|---|---|---|---|---|
| | Struc↑ | Comp↑ | Rec↑ | Prec↑ | wdist($\rho$)↓ | wdist($N_{el}$)↓ |
| **Crystal-LLM with LLaMA-2** | | | | | | |
| 70B ($\tau = 1.0$) | 0.983 | 0.892 | 0.979 | 0.984 | 1.63 | 0.53 |
| 70B ($\tau = 0.7$) | 0.997 | 0.971 | 0.873 | 0.988 | 0.77 | 0.42 |
| **MatExpert with LLaMA-2** | | | | | | |
| 70B ($\tau = 1.0$) | 0.991 | 0.913 | 0.986 | 0.992 | 0.18 | **0.08** |
| 70B ($\tau = 0.7$) | 0.998 | 0.981 | 0.982 | 0.994 | 0.21 | **0.08** |
| **MatExpert with LLaMA-3** | | | | | | |
| 70B ($\tau = 1.0$) | 0.996 | 0.922 | 0.991 | **0.997** | 0.15 | **0.08** |
| 70B ($\tau = 0.7$) | **0.999** | **0.986** | **0.989** | 0.985 | **0.14** | 0.09 |

**Crystal-LLM Gruver et al. (2024):** A state-of-the-art approach fine-tuning large language models (LLaMA-2 Touvron et al. (2023)) for generating stable inorganic materials. It excels in generating valid crystal structures and serves as a primary baseline due to its pioneering use of LLMs for material generation.

**CDVAE Xie et al. (2022b):** A Crystal Diffusion Variational Autoencoder tailored for periodic material generation. It combines generative models within a continuous VAE latent space, widely used for generating stable materials and considered a strong benchmark.

**LM-CH and LM-AC Flam-Shepherd & Aspuru-Guzik (2023):** Transformer-based models for 3D crystal structure generation. LM-CH uses character-level tokenization, while LM-AC uses atom and coordinate-level tokens.

For a fair comparison, MatExpert and baselines are trained on the same datasets for each experiment.

## 5.2 UNCONDITIONAL GENERATION

In the unconditional generation task, we aim to assess the ability of MatExpert to produce novel and stable material structures without any specific property constraints. This stage serves as a benchmark for evaluating the intrinsic generative capabilities of the model, focusing on the validity, diversity, and stability of the generated materials. For unconditional generation, we randomly select a material from the database during the first stage of MatExpert.

The comparison results are shown in Table 1. As evident from the table, the MatExpert family with LLaMA-3 (Dubey et al., 2024) outperforms all other methods across all metrics except metrics of coverage. Traditional models such as CDVAE, LM-CH, and LM-AC excel in coverage and structural validity. When compared to the state-of-the-art LLM-based model Crystal-LLM, our MatExpert method consistently outperforms Crystal-LLM across all settings, including the same model size and temperature as well as DFT-based stability. Notably, MatExpert achieves remarkable results in distribution metrics, benefiting from the transition from an existing material in the database. While Crystal-LLM may struggle with the hallucinations of LLM, thus generating out-of-distribution materials.

## 5.3 CONDITIONAL GENERATION

In the conditional generation task, we evaluate MatExpert's ability to generate material structures that meet specific property constraints using the large-scale NOMAD dataset. This task is crucial for practical applications, where researchers often require materials with targeted properties, such as specific band gaps, space groups, and chemical compositions, etc. As shown in Table 2, MatExpert significantly outperforms the baseline Crystal-LLM for all metrics. MatExpert also excels in property distribution, achieving much lower Wasserstein distances, which reflects its ability to generate materials with property distributions closely aligned with the training data. Notably, the stability metric like the energy above the hull

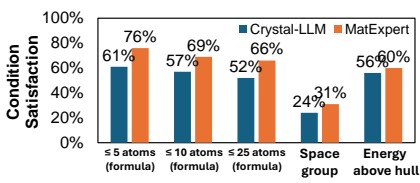

Figure 4: Conditional satisfaction rates of common property constraints. MatExpert consistently outperforms baseline methods.

is treated as an input property in this task and is not separately evaluated as a performance metric.

To rigorously evaluate the effectiveness of MatExpert in satisfying user-defined property constraints, we also conducted additional experiments aimed at quantifying the percentage of generated materials that meet specific property criteria set by users.

We selected a set of common property constraints, including:

- Formula: We parse the composition from the generated CIF. We show the results in three levels by atom numbers: $\leq 5$ atoms, $\leq 10$ atoms, and $\leq 15$ atoms.
- Energy above hull: We use M3GNet Chen & Ong (2022) to estimate the energy above hull. We bin stability as $\hat{E}^{hull} < 0.1$ for metastable and $\hat{E}^{hull} \geq 0.1$ for unstable.
- Space Group Number: We use PyMatgen's SpacegroupAnalyzer with a precision of 0.2 angstroms. Ong et al. (2013)

For each constraint, we generated 10,000 materials using MatExpert and the baselines. We then calculated the percentage of materials that met the specified property criteria. The results of the condition satisfaction experiments are summarized in Figure 4. MatExpert consistently outperformed the baselines across all property constraints, demonstrating its superior capability to generate materials that meet user-defined specifications.

## 6 ANALYSIS

### 6.1 ANALYSIS OF DIVERSITY AND NOVELTY

To evaluate the *MatExpert* framework, we use metrics that assess the diversity and novelty of generated materials. Following Xie et al. (2022b), diversity is calculated as the pairwise distance between samples using a featurization of structure and composition. Novelty is determined by comparing the distance to the nearest element of the training set for each sample, with a sample considered novel if this distance exceeds a threshold. We also assess the overall novelty, defined as having either a new structure or composition. To further understand the gaps between MatExpert and baselines, we illustrate the score of each metric normalized to testing samples in Figure 5. As we can see, MatExpert consistently achieves high-level scores in both structure and composition diversity compared to CDVAE and various Crystal-LLM configurations. This indicates MatExpert's superior ability

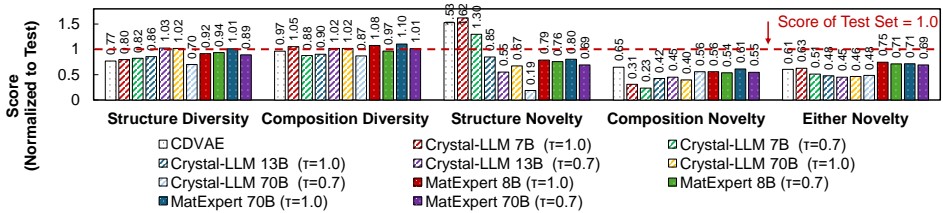

Figure 5: Scores of diversty and novelty normalized to testing samples. MatExpert consistently achieves remarkable scores on all metrics. All socres are calculated on the generated materials passed the validity.

to explore a broad chemical space. In terms of novelty, MatExpert demonstrates a strong capacity to generate structures and compositions not found in existing databases, as evidenced by its high novelty scores. Notably, Crystal-LLM faces challenges in balancing model size and novelty, with larger models exhibiting lower novelty. In contrast, MatExpert consistently maintains high novelty regardless of model size.

## 6.2 ABLATION STUDIES

To understand the contribution of different components of our framework, we conduct ablation studies by systematically removing or modifying key elements of MatExpert and observing the impact on performance.

**Impact of Transition (CoT) Stage** The transition stage, which involves generating modification pathways using Chain of Thought (CoT) reasoning Wei et al. (2022), plays a crucial role in the overall performance of MatExpert. To evaluate the impact of this component, we conducted experiments where the CoT reasoning was removed entirely. The results in Figure 6 highlight its effectiveness of CoT reasoning.

**Impact of Retrieval Stage** We assess the impact of the retrieval stage with the contrastive learning framework. By removing the retrieval stage, we observe a noticeable decline in the quality of the retrieved materials (See Figure 6). This confirms that the retrieval stage is crucial for identifying high-quality reference materials, which are essential for effective subsequent modifications.

**Choices of Encoders in Retrieval Stage** In the retrieval stage, we explored various encoder architectures to determine their impact on the quality of material retrieval (See Table 3). We compared T5-based encoders with BERT Devlin et al. (2019). Our findings indicate that T5-based encoders consistently outperform BERT, providing more accurate embeddings for both property and structure descriptions,

Table 3: Choices of encoders in retrieval stage. We use top-1 and top-5 accuracy, along with the average rank of the ground-truth materials as metrics.

| Encoder | Accuracy | | Avg Rank |
|---|---|---|---|
| | Top-1 (%) | Top-5 (%) | |
| Bert | 59.96 | 84.51 | 2.36 |
| T5 (ours) | **71.35** | **93.54** | **1.59** |

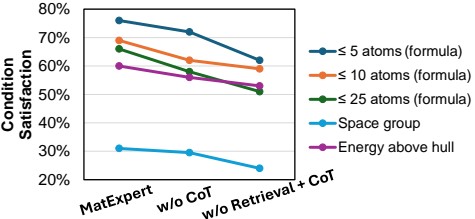

Figure 6: Impact of Transition (CoT) and Retrieval Stages. We sequentially remove the transition (CoT) stage and the retrieval stage to demonstrate changes in condition satisfaction rates.

making it more effective for capturing the most similar material aligned with the desired properties in the database.

## 6.3 CASE STUDY

To showcase the capabilities of MatExpert, we present a comprehensive example of conditional generation under property constraints in Figure 7. This case illustrates the process of generating a predicted material and its adherence to the specified properties.

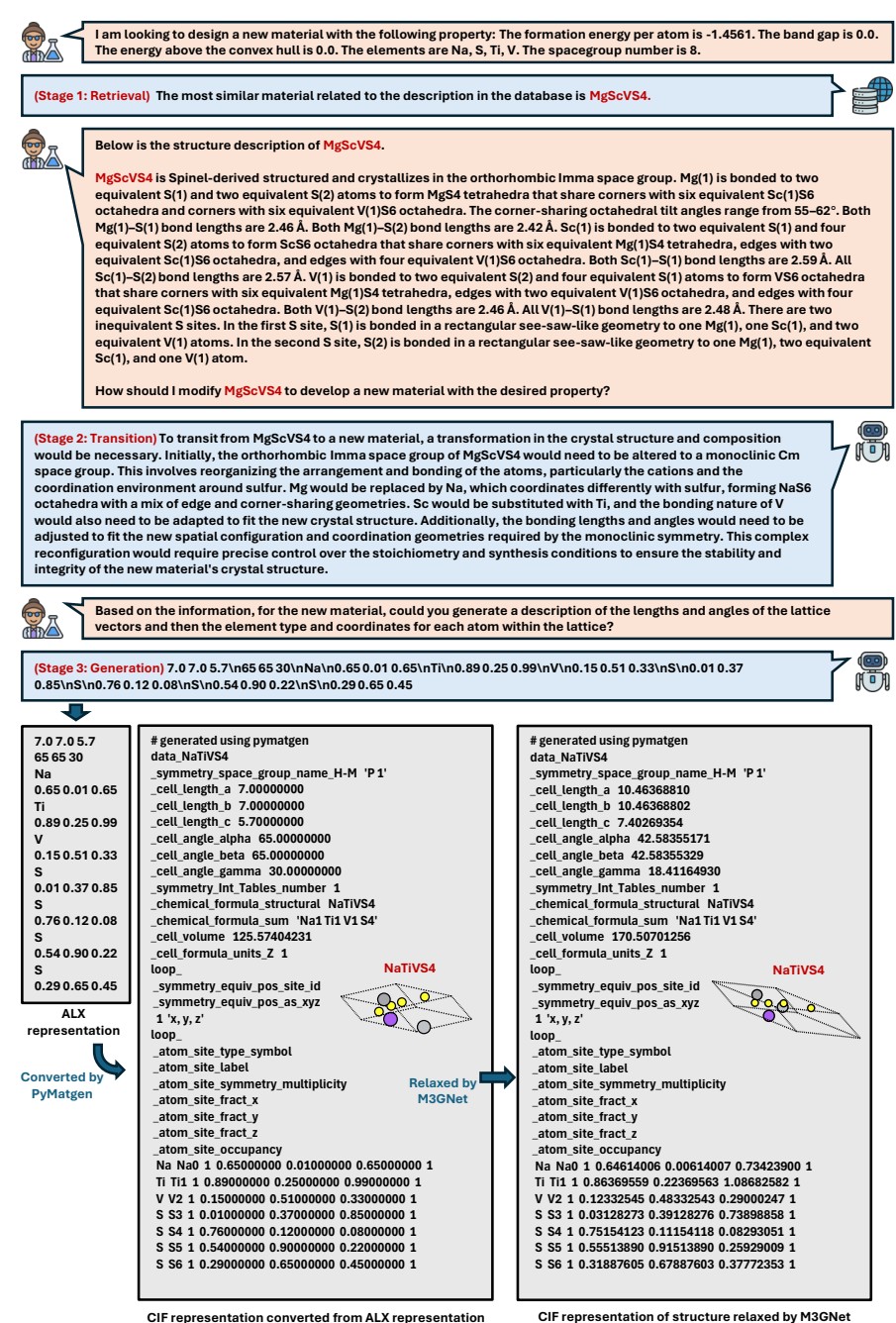

Figure 7: A comprehensive example of MatExpert for conditional generation: Enter the desired properties, and MatExpert will generate the CIF representation of the material.

## 7 CONCLUSION

In this work, we introduced MatExpert, a novel framework leveraging LLMs and contrastive learning for the material discovery process. By emulating the workflow of a human materials science expert, MatExpert integrates retrieval, transition, and generation stages to design new materials. Our experiments show that MatExpert significantly outperforms state-of-the-art methods in material generation tasks, thereby exhibiting its potential to become a scalable tool for accelerating material discovery with higher-quality crystal generation.

ACKNOWLEGMENTS

This work is supported by the Mila internal funding - Program P2-V1: Industry Sponsored Academic Labs (project number: 10379), the Canada CIFAR AI Chair Program, and the Canada NSERC Discovery Grant (RGPIN-2021-03115).

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

# A    REPRODUCIBLITY

## A.1    ENVIRONMENT

The experiments were conducted on machines with the following specifications.

Machine 1 for MatExpert with Llama-2 8B and Llama-3 8B:

- **Hardware:** NVIDIA A5000 GPU (24 GB on-board memory) GPU x 4, AMD Ryzen Threadripper PRO 3975WX CPU, 256 GB RAM
- **Software:** Ubuntu 22.04, Python 3.11, PyTorch 2.3.1+cu121, CUDA 12.4

Machine 2 for MatExpert with Llama-2 70B and Llama-3 70B:

- **Hardware:** NVIDIA L40S GPU (48 GB on-board memory) x 4, INTEL(R) XEON(R) GOLD 6526Y x 2, 512 GB RAM
- **Software:** Ubuntu 22.04, Python 3.11, PyTorch 2.3.1+cu121, CUDA 12.4

The list of environment requirements for the Python library will be made publicly available alongside the release of the open-source code of MatExpert.

## A.2    IMPLEMENTATION DETAILS

Table 4: Hyperparameters for training T5-based encoders in the retrieval stage.

| Hyperparameter | Value | Description |
|---|---|---|
| Model Architecture | t5-base | Pretrained model to use as the base encoder. |
| Learning Rate | 1e-4 | Learning rate for the optimizer. |
| Number of epochs | 100 | Number of epochs to train for. |
| Temperature | 0.1 | Temperature parameter for the NT-Xent loss. |
| Batch Size | 32 | Batch size for training. |
| Gradient Accumulation Steps | 8 | Number of batches for gradient accumulation. |

Table 5: Hyperparameters for finetuning Llama family in the transition and generation stage.

| Hyperparameter | Llama-3 8B | Llama-3 70B |
|---|---|---|
| Model Architecture | Meta-Llama-3-8B-Instruct | Meta-Llama-3-70B-Instruct |
| Learning Rate | $1e^{-4}$ | $1e^{-4}$ |
| Batch Size | 4 | 4 |
| Batch Size per Device | 1 | 1 |
| Number of Epochs | 70 | 10 |
| Optimizer | AdamW | AdamW |
| Sequence Length | 2048 | 1024 |
| LoRA Rank | 8 | 8 |
| LoRA Alpha | 16 | 16 |
| LoRA Dropout | 0.1 | 0.1 |
| Warmup Ratio | 0.1 | 0.1 |
| Gradient Accumulation Steps | 8 | 8 |

The proposed MatExprt framework consists of two training stages: 1) Training T5-based encoders of the contrastive learning framework in the stage Retrieval. 2) Finetuning the Llama-2/Llama-3 based LLMs in the stage Transition and stage Generation.

**Training T5-based encoders**    In the retrieval stage, two T5-based encoders are employed. The model uses the "t5-base" architecture in Huggingface Wolf (2019) as its foundational structure. The training process is governed by a learning rate of 0.0001. It undergoes training for 100 epochs with early stopping. A temperature parameter of 0.1 is utilized, which affects the NT-Xent loss function, a common choice for contrastive learning. The batch size is set to 32 and gradient accumulation occurs over 8 steps, effectively simulating a larger batch size and allowing for more stable updates.

**Finetuning Llama family**  In the transition and generation stages, the Llama family of models is finetuned. Two variants are used: Llama-3 8B and Llama-3 70B, both with specific architectures designed for instruction-based tasks named "Meta-Llama-3-8B-Instruct" and "Meta-Llama-3-70B-Instruct" in Huggingface Wolf (2019). The learning rate is set to 0.0001. Each device processes a batch size of 1, with an overall batch size of 4. The 8B model is trained for 70 epochs, while the 70B model undergoes 10 epochs, reflecting differences in their complexity and training needs. The AdamW optimizer is employed for parameter updates. Sequence lengths differ, with the 8B model handling sequences of 2048 tokens, and the 70B model processing 1024 tokens limited by the on-board memory of GPUs. LoRA settings include a rank of 8, an alpha of 16, and a dropout rate of 0.1, facilitating efficient adaptation. A warmup ratio of 0.1 helps stabilize initial training and gradient accumulation is set to 8 steps for both models, ensuring robust learning.

## A.3    EFFICIENCY

We utilize the open-source LLM training framework named LLaMA-Factory Zheng et al. (2024) to finetune the LLMs. With the advantages of LoRA Hu et al. (2022) and FlashAttention Dao et al. (2022) mechanism, it takes around 30 and 120 hours to fine-tune Llama-3 8B and Llama-3 70B in MatExpert, respectively.

The generation speed of MatExpert on our machines is as follows:

- MatExpert with Llama-3 8B: ∼40 minutes per 10,000 samples.
- MatExpert with Llama-3 70B: ∼5 hours per 10,000 samples.

## B    DATASETS DETAILS

Two datasets are included in our experiments: Material Project and NOMAD for unconditional generation and conditional generation, respectively.

**Material Project**  This dataset also named as MP-20 consisting of 45,231 materials is publicly available and was first proposed in Ong et al. (2013). We follow the data-split setting proposed in Gruver et al. (2023) and Xie et al. (2022a).

**NOMAD**  This large-scale dataset consisting of 2,886,120 materials is collected by ourselves and will be publicly available. We collect the datasets via NOMAD API as the following script:

```
url = 'http://nomad-lab.eu/prod/v1/api/v1/entries/archive/query'
excluded_elements = [
    "He", "Ne", "Ar", "Kr", "Xe", "Rn", "U", "Th", "Rn", "Tc",
    "Po", "Pu", "Pa",
]
query = {
    "results.method.simulation.program_name:any": [
        "VASP"
    ],
    "quantities:all": [
        "results.properties.structures",
        "results.properties.structures.structure_original",
        "results.properties.structures.structure_conventional",
        "results.properties.structures.structure_primitive"
    ]
}
required = {
    "results": {
        "material": {
            "chemical_formula_reduced": "*"
        },
        "properties": {
```

```
            "structures": "*"
        }
    }
}
```

We filter out the materials with elements in `excluded_elements` and only include the materials with structures computing by VASP Kresse & Hafner (1993).

Then we further convert the structure in JSON to CIF files via the following script:

```
def convert_to_cif(structure_json, lattice='vectors'):
    lattice_vectors = structure_json["lattice_vectors"]
    lattice_parameters = structure_json["lattice_parameters"]
    a = lattice_parameters["a"]
    b = lattice_parameters["b"]
    c = lattice_parameters["c"]
    alpha = lattice_parameters["alpha"] * 180 / np.pi
    beta = lattice_parameters["beta"] * 180 / np.pi
    gamma = lattice_parameters["gamma"] * 180 / np.pi

    if lattice == 'vectors':
        lattice = Lattice(lattice_vectors)
    elif lattice == 'parameters':
        lattice = Lattice.from_parameters(a, b, c, alpha,
                                          beta, gamma)

    positions = structure_json["cartesian_site_positions"]
    species = structure_json["species_at_sites"]

    structure = Structure(lattice, species,
                          positions, coords_are_cartesian=True)
    structure.scale_lattice(1000)

    return structure.to(fmt="cif")
```

We randomly select 10,000 samples and another 10,000 samples as the validation and testing set, respectively. The remaining samples are all included in the training set.

## C    EVALUATION METRICS

### C.1    VALIDITY

Validity is evaluated based on the structural and compositional integrity of the generated materials. Following Xie et al. (2022a), a structure is considered valid if the shortest distance between any pair of atoms is greater than 0.5 Å, ensuring atomic stability. Furthermore, the overall charge of the material must be neutral to satisfy chemical feasibility. These criteria ensure that the generated materials conform to realistic physical and chemical constraints.

### C.2    COVERAGE

Following the methods outlined in Xie et al. (2022a) and Gruver et al. (2023), we measure coverage using two metrics: COV-R (Recall) and COV-P (Precision). These metrics quantify how well the generated materials resemble the ground truth materials in the test set. COV-R (Recall) measures the percentage of ground truth materials accurately represented by the generated set, while COV-P (Precision) evaluates the quality of the generated materials in replicating the true material diversity. These metrics ensure that the generated materials not only capture the diversity of the test set but also maintain high-quality representations. Both validity and coverage are computed over 10,000 materials randomly sampled from the generated set.

## C.3 DISTRIBUTION

The distribution of properties is compared between the generated materials and the test set using the Wasserstein distance. The key properties evaluated are density ($\rho$, measured in g/cm$^3$) and the number of unique elements ($N_{el}$). The property distribution is computed over 1,000 valid materials, randomly sampled from those that pass the validity test. This method ensures that the generated materials exhibit realistic physical properties comparable to those in the test set.

## C.4 DIVERSITY

Diversity is assessed by calculating the pairwise distances between samples, based on structural and compositional features as described by Court et al. (2020). This metric quantifies how distinct each generated sample is relative to others in the dataset, offering insights into the variety of structures and compositions. Diversity is specifically measured on samples deemed metastable by M3GNet, as these are more likely to contribute meaningful variation. All diversity values are normalized against corresponding metrics from the test set, providing a clear comparison of the underlying data distributions.

## C.5 NOVELTY

Novelty is calculated by comparing each generated sample to its nearest neighbor in the training set. A sample is considered novel if its nearest neighbor exceeds a predefined threshold distance. Specifically, we use a structural distance cutoff of 0.1 and a compositional distance cutoff of 2. Novelty is assessed both in terms of structure and composition, with overall crystal novelty determined by the presence of a new structure or composition. These metrics provide insights into the uniqueness of the generated samples, particularly those identified as metastable. Novelty values are normalized by corresponding values for the test set to effectively convey the characteristics of the data distribution.

## C.6 STABILITY

We assess the stability of materials using a combination of machine learning potentials and density functional theory (DFT) to ensure a consistent evaluation framework. Specifically, we employ the M3GNet model Chen & Ong (2022), trained on total energy data from VASP calculations within the Materials Project dataset. This aligns the results with established correction schemes and absolute energy values. For consistency with Materials Project settings and prior works Xie et al. (2022a); Gruver et al. (2023), including the PBE functional and DFT/DFT+U, we perform a single relaxation for each candidate structure using the default MPRelaSet parameters.

To determine the percentage of metastable compounds, we first filter out samples that fail basic structural and compositional validity checks. The remaining samples are then relaxed using M3GNet to obtain final relaxation energies. The stability calculation includes the validity rate from initial filtering and the rate of compounds with relaxed hull energy $\hat{E}_{\text{hull}} < 0.1$. For stable materials, those identified as metastable by M3GNet undergo further DFT relaxation, and the percentage with $\hat{E}_{\text{hull}} < 0.0$ is reported. This comprehensive approach integrates both machine learning and traditional DFT methods for a robust stability evaluation.

## D CASES OF UNCONDITIONAL GENERATION

Please see Figure 8.

## E CASES OF CONDITIONAL GENERATION

Please see Figure 9.

## F NUMERICAL RESULTS OF ABLATION STUDY (FIGURE 6)

Please see Table 6.

```
3.7 3.7 5.6
90 90 119
U
0.78 0.28 0.10
U
0.44 0.61 0.63
P
0.11 0.95 0.35
N
0.78 0.28 0.74
N
0.44 0.61 0.98
```

```
# generated using pymatgen
data_U2PN2
_symmetry_space_group_name_H-M  'P 1'
_cell_length_a  3.70000000
_cell_length_b  3.70000000
_cell_length_c  5.60000000
_cell_angle_alpha  90.00000000
_cell_angle_beta  90.00000000
_cell_angle_gamma  119.00000000
_symmetry_Int_Tables_number  1
_chemical_formula_structural  U2PN2
_chemical_formula_sum  'U2 P1 N2'
_cell_volume  67.05184523
_cell_formula_units_Z  1
loop_
 _symmetry_equiv_pos_site_id
 _symmetry_equiv_pos_as_xyz
 1 'x, y, z'
loop_
 _atom_site_type_symbol
 _atom_site_label
 _atom_site_symmetry_multiplicity
 _atom_site_fract_x
 _atom_site_fract_y
 _atom_site_fract_z
 _atom_site_occupancy
 U U0 1 0.78000000 0.28000000 0.10000000 1
 U U1 1 0.44000000 0.61000000 0.63000000 1
 P P2 1 0.11000000 0.95000000 0.35000000 1
 N N3 1 0.78000000 0.28000000 0.74000000 1
 N N4 1 0.44000000 0.61000000 0.98000000 1
```

```
4.8 4.8 4.8
90 90 90
Si
0.76 0.62 0.92
C
0.26 0.12 0.92
C
0.26 0.62 0.42
N
0.26 0.12 0.42
N
0.26 0.62 0.92
N
0.76 0.12 0.42
N
0.76 0.62 0.42
```

```
# generated using pymatgen
data_Si(CN2)2
_symmetry_space_group_name_H-M  'P 1'
_cell_length_a  4.80000000
_cell_length_b  4.80000000
_cell_length_c  4.80000000
_cell_angle_alpha  90.00000000
_cell_angle_beta  90.00000000
_cell_angle_gamma  90.00000000
_symmetry_Int_Tables_number  1
_chemical_formula_structural  Si(CN2)2
_chemical_formula_sum  'Si1 C2 N4'
_cell_volume  110.59200000
_cell_formula_units_Z  1
loop_
 _symmetry_equiv_pos_site_id
 _symmetry_equiv_pos_as_xyz
 1 'x, y, z'
loop_
 _atom_site_type_symbol
 _atom_site_label
 _atom_site_symmetry_multiplicity
 _atom_site_fract_x
 _atom_site_fract_y
 _atom_site_fract_z
 _atom_site_occupancy
 Si Si0 1 0.76000000 0.62000000 0.92000000 1
 C C1 1 0.26000000 0.12000000 0.92000000 1
 C C2 1 0.26000000 0.62000000 0.42000000 1
 N N3 1 0.26000000 0.12000000 0.42000000 1
 N N4 1 0.26000000 0.62000000 0.92000000 1
 N N5 1 0.76000000 0.12000000 0.42000000 1
 N N6 1 0.76000000 0.62000000 0.42000000 1
```

```
4.9 4.9 9.3
90 90 90
Li
0.89 0.57 0.70
Li
0.39 0.07 0.20
Li
0.89 0.57 0.20
Li
0.39 0.07 0.70
Ge
0.39 0.07 0.95
Ge
0.89 0.57 0.45
F
0.09 0.88 0.95
F
0.09 0.88 0.45
F
0.18 0.38 0.95
F
0.18 0.38 0.45
F
0.59 0.27 0.62
F
0.68 0.77 0.12
F
0.68 0.77 0.62
F
0.59 0.27 0.12
F
0.59 0.77 0.95
F
0.68 0.27 0.45
F
0.09 0.27 0.95
F
0.18 0.77 0.45
```

```
# generated using pymatgen
data_Li2GeF6
_symmetry_space_group_name_H-M  'P 1'
_cell_length_a  4.90000000
_cell_length_b  4.90000000
_cell_length_c  9.30000000
_cell_angle_alpha  90.00000000
_cell_angle_beta  90.00000000
_cell_angle_gamma  90.00000000
_symmetry_Int_Tables_number  1
_chemical_formula_structural  Li2GeF6
_chemical_formula_sum  'Li4 Ge2 F12'
_cell_volume  223.29300000
_cell_formula_units_Z  2
loop_
 _symmetry_equiv_pos_site_id
 _symmetry_equiv_pos_as_xyz
 1 'x, y, z'
loop_
 _atom_site_type_symbol
 _atom_site_label
 _atom_site_symmetry_multiplicity
 _atom_site_fract_x
 _atom_site_fract_y
 _atom_site_fract_z
 _atom_site_occupancy
 Li Li0 1 0.89000000 0.57000000 0.70000000 1
 Li Li1 1 0.39000000 0.07000000 0.20000000 1
 Li Li2 1 0.89000000 0.57000000 0.20000000 1
 Li Li3 1 0.39000000 0.07000000 0.70000000 1
 Ge Ge4 1 0.39000000 0.07000000 0.95000000 1
 Ge Ge5 1 0.89000000 0.57000000 0.45000000 1
 F F6 1 0.09000000 0.88000000 0.95000000 1
 F F7 1 0.09000000 0.88000000 0.45000000 1
 F F8 1 0.18000000 0.38000000 0.95000000 1
 F F9 1 0.18000000 0.38000000 0.45000000 1
 F F10 1 0.59000000 0.27000000 0.62000000 1
 F F11 1 0.68000000 0.77000000 0.12000000 1
 F F12 1 0.68000000 0.77000000 0.62000000 1
 F F13 1 0.59000000 0.27000000 0.12000000 1
 F F14 1 0.59000000 0.77000000 0.95000000 1
 F F15 1 0.68000000 0.27000000 0.45000000 1
 F F16 1 0.09000000 0.27000000 0.95000000 1
 F F17 1 0.18000000 0.77000000 0.45000000 1
```

Figure 8: Cases of unconditional generation.

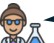

I am looking to design a new material with the following property: The formation energy per atom is -0.168. The band gap is 0.0. The energy above the convex hull is 0.0189. The elements are Hg, Li. The spacegroup number is 221.

```
4.3 4.3 4.3
90 90 90
Li
0.70 0.86 0.21
Hg
0.20 0.36 0.21
Hg
0.20 0.86 0.71
Hg
0.70 0.36 0.71
```

```
# generated using pymatgen
data_LiHg3
_symmetry_space_group_name_H-M  'P 1'
_cell_length_a  4.30000000
_cell_length_b  4.30000000
_cell_length_c  4.30000000
_cell_angle_alpha  90.00000000
_cell_angle_beta  90.00000000
_cell_angle_gamma  90.00000000
_symmetry_Int_Tables_number  1
_chemical_formula_structural  LiHg3
_chemical_formula_sum  'Li1 Hg3'
_cell_volume  79.50700000
_cell_formula_units_Z  1
loop_
 _symmetry_equiv_pos_site_id
 _symmetry_equiv_pos_as_xyz
 1 'x, y, z'
loop_
 _atom_site_type_symbol
 _atom_site_label
 _atom_site_symmetry_multiplicity
 _atom_site_fract_x
 _atom_site_fract_y
 _atom_site_fract_z
 _atom_site_occupancy
 Li Li0 1 0.70000000 0.86000000 0.21000000 1
 Hg Hg1 1 0.20000000 0.36000000 0.21000000 1
 Hg Hg2 1 0.20000000 0.86000000 0.71000000 1
 Hg Hg3 1 0.70000000 0.36000000 0.71000000 1
```

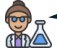

I am looking to design a new material with the following property: The formation energy per atom is -0.0431. The band gap is 0.0. The energy above the convex hull is 0.0531. The elements are Mn, Ge. The spacegroup number is 194.

```
4.0 4.0 5.1
90 90 120
Mn
0.67 0.12 0.56
Mn
0.67 0.12 0.06
Mn
0.00 0.78 0.31
Mn
0.33 0.45 0.81
Ge
0.33 0.45 0.31
Ge
0.00 0.78 0.81
```

```
# generated using pymatgen
data_Mn2Ge
_symmetry_space_group_name_H-M  'P 1'
_cell_length_a  4.00000000
_cell_length_b  4.00000000
_cell_length_c  5.10000000
_cell_angle_alpha  90.00000000
_cell_angle_beta  90.00000000
_cell_angle_gamma  120.00000000
_symmetry_Int_Tables_number  1
_chemical_formula_structural  Mn2Ge
_chemical_formula_sum  'Mn4 Ge2'
_cell_volume  70.66767295
_cell_formula_units_Z  2
loop_
 _symmetry_equiv_pos_site_id
 _symmetry_equiv_pos_as_xyz
 1 'x, y, z'
loop_
 _atom_site_type_symbol
 _atom_site_label
 _atom_site_symmetry_multiplicity
 _atom_site_fract_x
 _atom_site_fract_y
 _atom_site_fract_z
 _atom_site_occupancy
 Mn Mn0 1 0.67000000 0.12000000 0.56000000 1
 Mn Mn1 1 0.67000000 0.12000000 0.06000000 1
 Mn Mn2 1 0.00000000 0.78000000 0.31000000 1
 Mn Mn3 1 0.33000000 0.45000000 0.81000000 1
 Ge Ge4 1 0.33000000 0.45000000 0.31000000 1
 Ge Ge5 1 0.00000000 0.78000000 0.81000000 1
```

Figure 9: Cases of conditional generation.

Table 6: Numerical results of Ablation Study (Figure 6)

| MatExpert | - CoT Stage | - CoT Stage | - Retreval Stage |
|---|---|---|---|
| $\leq$ 5 atoms | 76% | 72% | 62% |
| $\leq$ 10 atoms | 69% | 62% | 59% |
| $\leq$ 25 atoms | 66% | 58% | 51% |
| Space group | 31% | 30% | 24% |
| Energy above hull | 60% | 56% | 53% |

