# OpenReview forum: "MatExpert: Decomposing Materials Discovery By Mimicking Human Experts"
_ICLR.cc/2025/Conference — ICLR 2025 Poster_

### Official Review · Reviewer_2LRH · 2024-10-30

**Soundness:** 2
**Presentation:** 2
**Contribution:** 2
**Rating:** 6
**Confidence:** 3

**Summary:**

The paper proposes MatExpert framework, designed to enhance LLM-based crystal structure generation through a 3-stage inference. The first stage searches an existing structure that closely resembles the user's input via T5 model trained with a contrastive learning objective on property and structure descriptions. Then, the fine-tuned LLM suggests modifications to the physical properties and structural attributes to achieve the target composition and characteristics. Finally, the LLM generates an ALX representation, which is converted to the final crystal structure. The authors also introduce a curated, large-scale dataset for training and demonstrate that this iterative, feedback-driven approach improves all evaluation metrics.

**Strengths:**

1. This paper presents a novel approach by modeling crystal structures with a chain-of-experts framework that utilizes multiple LLMs. As far as I know, this approach is novel, and appears effective in generating stable materials.
2. Additionally, the use of fine-tuning methods, such as LoRA and distillation, enhances the framework’s efficiency and scalability, making it practical for real-world adaptation.

**Weaknesses:**

1. The authors mentioned in the paper that “In the unconditional generation task, we aim to assess the ability of MatExpert to produce novel and stable material structures without any specific property constraints”, while at the same time “... For unconditional generation, we randomly select a material from the database during the first stage of MatExpert.” According to the second referred sentence, the second, transition stage would take samples of training set embeddings or raw data would be given as input. This is contradictory to the claim that no structure or property is given to generate. For this reason, I believe the evaluation scheme is unfair. Please fix me in the author response if I understood incorrectly.

2.  Following this concern, the performance improvement on generation is also questionable. The authors are encouraged to provide stabilities (Predicted energy over convex hull values, or DFT relaxation success rate) of the generated samples that are out-of-distribution. Also, stability measurements on Table 2 are also needed.

3. There are stronger baselines following the CDVAE research, as the authors stated. However, comparisons between them, which is necessary, are missing in Table 1.  Furthermore, the authors also need to specify which dataset was used, or, the source of the CDVAE model. If the CDVAE model is trained only on MP-20, that would not be a fair comparison.

**Questions:**

1. In Figure 5, are all the structures relaxed or not? Does it contain all generated samples, or samples that passed the validity test? Please provide the details.

2. Have you checked the energy value difference between the model inputs and generated samples? If there are OOD samples, does it pass the validity test or relaxation steps?

3. Regarding Equation (1), what do you use as the similarity function?

4. What prompts have you given to the model? Concretely, how much overlap is there between the conditioned properties and the NOMAD data?

---

> ### Author Response · Authors · 2024-11-20
>
> We appreciate the valuable feedback and have carefully considered the points raised. Below, we address your concerns. We hope our revisions clarify all the questions and strengthen the quality of our work.
>
> > **W1** Unconditional Generation Task Clarification
>
> **AW1** We apologize for any confusion regarding the definition of “unconditional” in our comparison.
>
> To clarify:
>
> **Misunderstanding.** It seems there might be a misunderstanding regarding our use of the term "unconditional." In our context, "unconditional" refers to generating materials without any predefined constraints on properties such as bandgap, space group, or composition. The goal is to create stable materials that have not been seen in the database.
>
> **No contradictory.** The retrieval stage in MatExpert operates as a retrieval-augmented-generation (RAG) technique, where we retrieve materials from the training set. This approach ensures that no new information or knowledge beyond what is available in the training data is introduced. The seed structure is selected randomly from the training set and does not impose any constraints, ensuring a fair comparison with other baseline methods. We will clarify this distinction in the revised manuscript to address any potential misunderstandings.
>
> > **W2** Performance Improvement and Stability Measurements
>
> **AW2** We understand the importance of demonstrating the stability of out-of-distribution (OOD) generated samples. Here is how we consider this issue:
>
> We define anything new in the generated materials as out-of-distribution (OOD). For the "unconditional generation" task, we have already calculated stabilities for all materials generated by MatExpert and the baseline models, including stability and metastability metrics. This calculation covers all generated materials that passed the validity tests.
>
> There are two phases in our stability calculations:
>
> **Pre-relaxation with M3GNet:** We initially estimate stability using M3GNet predictions.
>
> **Correction by DFT Relaxation:** We further refine these estimates through relaxation with DFT.
>
> This two-step approach ensures that our stability assessments are comprehensive and align with standard practices, providing a fair comparison with other baselines.
>
> For the definitions of stability and metastability, please refer to Appendix C.6. For meta-stability, the calculation was applied to the generated materials that passed the validity test, which indicates the rate of compounds with relaxed hull energy $\hat{E}\_{\text{hull}} < 0.1$ by M3GNet. For stability, those identified as metastable by M3GNet undergo further DFT relaxation, and we report the percentage with $\hat{E}_{\text{hull}} < 0.0$.
>
> Regarding "conditional generation," the convex hull of NOMAD is not publicly available, so we lack reference energy data for NOMAD materials. Due to computational resource limitations, it is not feasible to calculate the convex hull by ourselves for over 2 million materials in NOMAD. Therefore, we only focus on the performance of the generated materials that meet user-defined property constraints for "conditional generation”.

---

> > ### Author Response · Authors · 2024-11-20
> >
> > > **W3** Comparison with Stronger Baselines
> >
> > **AW3** We appreciate the suggestion to include a more comprehensive comparison with stronger CDVAE family models, such as DiffCSP and UniMat.
> >
> > To clarify:
> >
> > **Fair comparison.** The results for CDVAE are derived from the official CDVAE model trained on the MP-20 dataset. MatExpert for unconditional generation was also trained exclusively on the MP-20 dataset as all other baselines. For conditional generation, MatExpert and baselines were all trained exclusively on the NOMAD dataset. We did not train MatExpert on both datasets simultaneously, ensuring a fair and consistent comparison across models. The clarification is now further included in line 316, line 321, and line 370 in the updated manuscript.
> >
> > **Comparison.** The below comparison table highlights the performance of MatExpert against diffusion-based models DiffCSP[1] and UniMat[2]. It is important to note that CDVAE, DiffCSP[1], and UniMat[2] are diffusion-based and all designed only for "unconditional generation" and do not support inputting constraints on properties. This limitation highlights MatExpert's advantage in handling both unconditional and conditional generation tasks, providing broader applicability and flexibility in material discovery.
> >
> > || Validity|| Coverage | | Distribution ||
> > |--|--|--|--|--|--|---|
> > || Struc.  | Comp.  | Rec.     | Pre.     | wdist $\rho$   | wdist $N_{el}$ |
> > | DiffCSP   | 100    | 83.25    | 99.71    | 99.76    | 0.35         | 0.33       |
> > | UniMat  | 97.2    | 89.4     | **99.8** | **99.7** | **0.09**     | 0.06       |
> > | MatExpert 70B (\tau=0.7) | **99.8**  | **96.1** | 98.6     | 99.1     | 0.18         | **0.04**   |
> >
> > [1] Crystal Structure Prediction by Joint Equivariant Diffusion, Jiao, Rui et al., NeurIPS 2023
> >
> > [2] UniMat: Scalable Diffusion for Materials Generation, Sherry Yang et al.
> >
> > **Reproducibility.** UniMat's source code and weights are not publicly available, preventing a direct comparison in efficiency. However, based on the computational costs stated in UniMat’s paper “utilizing 32 TPUv4 units for training”, UniMat's resource requirements are significantly higher than MatExpert, indicating that our approach is more computationally efficient.
> >
> > > **Q1** Figure 5 Details
> >
> > **AQ1** Thanks for pointing out this confusion. In Figure 5, we followed the settings used in CrystaLLM, where all structures were considered before being relaxed to calculate these metrics. The metrics are calculated only on the generated samples that passed the validity test.
> >
> > > **Q2** Energy Value Differences and OOD Samples
> >
> > **AQ2** We apologize for this confusion.
> >
> > To clarify:
> >
> > The MP-20 dataset contains over 40,000 materials, making it computationally prohibitive to calculate the energy above the hull using DFT for every training sample due to the high expense of these calculations. We aim to address this in the future as we have more time to run such experiments.
> >
> > This manuscript focuses on material discovery, so all generated materials are considered out-of-distribution (OOD) samples. Figure 5 demonstrates that the novelty scores for the MatExpert group are significantly higher than those of the baselines, indicating its strong capability to generate OOD materials. The definition of novelty is stated in Appendix C.5.
> >
> > We calculated all metrics using the generated materials that passed the validity tests, and we only employed relaxation for stability calculations.
> >
> > > **Q3** Similarity Function in Equation (1)
> >
> > **AQ3** Thanks for pointing out this confusion. We use cosine similarity as the similarity function in Equation (1). This choice effectively measures the alignment between structure and property embeddings. This is now included in Line 201  in the updated manuscript.
> >
> > > **Q4** Prompts and Overlap with NOMAD Data
> >
> > **AQ4** The prompts provided to the model are constructed from a comprehensive range of properties, including all properties extracted by Robocrystallographer from the NOMAD data. This ensures broad applicability and flexibility. During the training stage, we randomly select properties to construct our prompts, allowing adaptability to various conditions. The set of conditions is fixed only during the evaluation stage to assess specific properties.
> >
> > Prompt template for reference:
> >
> > ```
> > I am looking to design a new material with the following property: <property_1> is/are <property_value_1>. <property_2> is/are <property_value_2>. <property_3> is/are >property_value_3>...  The closest existing material I found in the database is <formula source>, which has similar properties. Below is the structure description of <formula source>.
> >
> > <description_source extracted from Robocrystallographer>
> >
> > How should I modify <formula_source> to develop a new material with the desired properties?
> > ```

---

> ### Comment · Reviewer_2LRH · 2024-12-01
>
> I appreciate the reviewer for the detailed responses.
>
> My concerns have been partially addressed. Regarding W3, the authors provided experimental evidence showing that the proposed method is more stable than diffusion-based models, and it seems it is because of the RAG mentioned in W1.
>
> However, it is still questionable for me how the authors conducted unconditional generation. The provided prompts are unclear if they are conditional or unconditional. Other reviewers have commented similar aspects, but I am not fully persuaded by the rebuttal.
>
> For this reason, I'd like to raise the score to 4, but the system does not offer the rating of 4, so I am just leaving this comment here.

---

> > ### Author Response · Authors · 2024-12-01
> >
> > Thank you for raising your concerns about conditional and unconditional generation. And we are grateful for your consideration in raising the score. We appreciate the opportunity to clarify this aspect as follows:
> >
> > **Unconditional Generation:** The unconditional generation was conducted exclusively on the MP-20 dataset. For this mode, there are **NO** pre-set prompts. Instead, we skip the prompt and randomly select a material from the database. The system then generates a pathway to modify this randomly selected material into a novel material, with no specific property constraints guiding the process.
> >
> > **Conditional Generation:** In contrast, conditional generation was conducted using the NOMAD dataset, where user-specified property constraints (such as desired bandgap, formation energy, or space group) were applied. The prompts for conditional generation explicitly incorporate these property constraints to guide the model in retrieving reference materials and generating new materials that meet the specified criteria.
> >
> > In Q4, you specifically requested example prompts for the NOMAD dataset, so we provided example prompts only for conditional generation, which incorporates user-defined property constraints.
> >
> > We hope this clarification addresses your concern regarding the setting of conditional and unconditional generation. Please let us know if this response resolves your questions or if further clarification is needed.

---

> > > ### Comment · Reviewer_2LRH · 2024-12-02
> > >
> > > Thank you for the detailed clarification regarding the unconditional and conditional generation processes. This resolves my previous questions and concerns about the experimental setup. Given this explanation and the earlier evidence provided about the stability compared to diffusion-based models, I am happy to raise the score to 6. Thanks!

---

> > > > ### Author Response · Authors · 2024-12-02
> > > >
> > > > Thank you for your positive feedback and for taking the time to review our manuscript thoroughly. We are delighted to hear that the detailed clarification regarding the unconditional and conditional generation processes effectively addressed your concerns. We also appreciate your acknowledgment of the stability of our approach compared to diffusion-based models.
> > > >
> > > > We are grateful for your decision to raise the score of our submission. We are committed to continuing to improve our research and are encouraged by your thoughtful insights. Thank you once again for your consideration.

---

### Official Review · Reviewer_SCz7 · 2024-11-01

**Soundness:** 2
**Presentation:** 3
**Contribution:** 3
**Rating:** 6
**Confidence:** 4

**Summary:**

The paper presents MatExpert, a framework that leverages large language models (LLMs) and contrastive learning to automate materials discovery. The proposed approach mimics human experts by breaking the process into three stages: retrieval, transition, and generation. MatExpert first identifies a closely related material, then suggests modifications to meet target specifications, and finally generates new material structures. The authors test MatExpert with large datasets from NOMAD and the Materials Project, showing that it performs better than current methods on key measures like stability, validity, and how well it meets diversity and novelty.

**Strengths:**

* The integration of Robocrystallographer enriches crystal data with textual descriptions, enhancing the retrieval process and interpretability.
* MatExpert achieves impressive performance on benchmarks, demonstrating its reliability in generating valid and diverse material structures.
* Contrastive learning effectively maps structure and property embeddings, which is a novel approach for aligning multimodal material data.

**Weaknesses:**

* The novelty of multi-stages material generation is a bit limited as it’s being studied in other works [1, 2, 3].  In the introduction, the author mention the drawback of the current method is the single step material structure generation. However, some cited paper include multi-steps material generation and property query already [1, 2, 3]. It will be helpful to have more discussion on those methods.
* The paper could benefit from more clarity on the pathway generation process. Specifically, it is unclear how the pathways generated by GPT-4 can be reliably reproduced in real-world lab settings. The authors might find [4] useful as a reference for evaluating the quality and safety of generated pathways.
* Due to the complexity of multi-steps framework, the paper could discuss more on how the authors prevent the error propagation.
* The visualization for Figure 6 is not clear enough. If there’s a table include the numerical value of the ablation study, it can better show the improvement of each component.

**Questions:**

* Follow up on W1, given the reproducibility challenges in LLM-generated content, how does the framework handle multiple potential pathways for synthesizing a target material?
* Follow up on W3, I wonder what’s MatExpert’s the success rate on each step? Specifically, the success rate for generating accurate ALX representation.
* Can this method be useful for user query without specifying formation energy and band gap for the target material? For instance, can the user prompt the model like, I want to material composed of Mn, Ge and with high electrical conductivity.
* What will the model response if the target material doesn’t exist?

[1] Miret, Santiago, et al. “Are LLMs Ready for Real-World Materials Discovery?”

[2] Zhang, Huan, et al. “HoneyComb: A Flexible LLM-Based Agent System for Materials Science”

[3] Chiang, Yuan, et al. “LLaMP: Large Language Model Made Powerful for High-fidelity Materials Knowledge Retrieval and Distillation”

[4] Microsoft Research AI4Science. “The Impact of Large Language Models on Scientific Discovery: a Preliminary Study using GPT-4”

---

> ### Author Response · Authors · 2024-11-20
>
> We appreciate the valuable feedback and have carefully considered the points raised. Below, we address your concerns. We hope our revisions clarify all the questions and strengthen the quality of our work.
>
> > **W1** Novelty of Multi-Stage Generation
>
> **AW1** We appreciate your references to other works [1, 2, 3]. However, our contribution focuses on novel material discovery, which is distinctly different from the tasks addressed in those papers. Specifically, [1] is a positioning framework rather than a specific method; [2, 3] propose retrieval-augmented generation (RAG) methods for textual documents in material science. our approach introduces a new paradigm of retrieval, refinement, and generation for conditional material generation, leveraging material structures rather than NLP corpora.
>
> [1] Miret, Santiago, et al. “Are LLMs Ready for Real-World Materials Discovery?”
>
> [2] Zhang, Huan, et al. “HoneyComb: A Flexible LLM-Based Agent System for Materials Science”
>
> [3] Chiang, Yuan, et al. “LLaMP: Large Language Model Made Powerful for High-fidelity Materials Knowledge Retrieval and Distillation”
>
>
> > The novelty of our work can be divided into the following points:
>
> **Novel Paradigm:** Our framework introduces a new paradigm for material discovery by integrating retrieval, refinement, and generation stages. Specifically in the context of conditional material generation, MatExpert demonstrates remarkable flexibility and ability to generate a novel material conditioning on the user-predefined constraints on properties with this new paradigm and finetuned LLMs. To the best of our knowledge, it is the first to work with this paradigm and demonstrate effectiveness on large-scale material benchmark datasets.
>
> **Novel Retrieval Approach:** Our retrieval approach, the first step of MatExpert, is different from traditional retrieval-augmented generation (RAG) approaches commonly used in natural language processing (NLP). Traditional RAG methods typically focus on retrieving relevant text snippets or knowledge pieces from large NLP corpora to enhance the generation process. In contrast, our approach is tailored specifically for material generation, where we retrieve material structures from databases instead of textual information. This structural retrieval serves as a seed for subsequent modifications, providing a tangible and scientifically grounded starting point for material generation.
>
> **Novel Cain-of-Thought Approach:** In our framework, we use a Chain-of-Thought (CoT) reasoning approach to enhance the transition stage. GPT-4 takes both the true material structure (answer) and the user query as inputs to generate detailed modification steps (thought). These steps serve as ground-truth pathways for fine-tuning LLaMA. LLaMA, in turn, takes the query alone as input and is trained to generate both the modification pathways (thought) and the resulting material structure (answer). This approach enables LLaMA to apply expert-like reasoning, ensuring accurate and effective material modifications. With the inclusion of contrastive learning and CoT reasoning, our framework provides interpretable pathways for material modifications, allowing users to evaluate the reliability and reasonableness of generated outcomes.
>
> **Dataset Innovation:** Our work collects and leverages the NOMAD database, comprising over 2.8 million materials, to set a new standard in material generation models. This extensive dataset, larger and more diverse than those typically used, like MP-20 or Carbon-24, reduces the risk of overfitting and enhances model generalization. Our approach not only improves material generation performance but also encourages the use of large-scale, high-quality datasets for more robust discoveries in materials science.

---

> > ### Author Response · Authors · 2024-11-20
> >
> > > **W3** Error Propagation in Multi-Steps Framework
> >
> > **AW3** We acknowledge the concern regarding error propagation in the multi-stage process. Here’s how we consider this issue:
> >
> > **Only limited rounds:** Since the process of MatExpert involves only two rounds of conversation with one step of refinement, the potential for error accumulation is minimized. Our empirical results show that despite taking the risk of error propagation, the overall performance improvements justify the multi-stage approach.
> >
> > **Further Improvement:** With additional computational resources, we can have the following improvement directions: 1) implement feedback loops to allow for iterative refinements, thus addressing any initial inaccuracies in the retrieval stage. 2) With tools like pymatgen and m3gnet, we can check whether the retrieved material at the initial step is similar to the requested property constraints to avoid error propagation. Additionally, we only use a very trivial retrieval approach with only one step of refinement to demonstrate the effectiveness of the multi-stage process, we believe with the most advanced RAG technique, it will be further improved.
> >
> > > **W4** Visualization in Figure 6
> >
> > **AW4** Thank you for pointing out the issue with Figure 6. Our initial intention with Figure 6 was to use a line chart to illustrate the trend of performance degradation as the CoT and Retrieval modules are progressively removed. This visualization aims to highlight the contribution and importance of each module within our framework. We included a table with numerical values in Appendix F to clearly illustrate the impact of each component:
> >
> > |                   | MatExpert | - CoT Stage  | - CoT Stage      - Retreval Stage |
> > |-------------------|-----------|--------------|-----------------------------------|
> > | ≤ 5 atoms         | 76%       | 72%          | 62%                               |
> > | ≤ 10 atoms        | 69%       | 62%          | 59%                               |
> > | ≤ 25 atoms        | 66%       | 58%          | 51%                               |
> > | Space group       | 31%       | 30%          | 24%                               |
> > | Energy above hull | 60%       | 56%          | 53%                               |
> >
> > > **Q1** Handling Multiple Potential Pathways
> >
> > **AQ1** As a large language model, MatExpert is designed to predict a single potential pathway by default. However, we can enable the model to generate multiple pathways by running multiple inference rounds. This approach allows for the exploration of various material candidates that potentially satisfy the given conditions. Note that this issue is a natural challenge for all baselines and state-of-the-art methods in material generation.
> >
> > As we mentioned in **AW2**, we do not propose synthesis pathways for laboratory use. We appreciate your question and we will explore how to align LLM-generated pathways more closely with real-world synthesis processes.
> >
> > > **Q2** Success Rate of Each Step
> >
> > **AQ2** We have evaluated the success rate for generating accurate ALX and CIF representations. Preliminary results on MatExpert-8B indicate high success rates in format, with 99.6% for ALX representation and 96.8% for CIF representation, which reflect the validity for both ALX and CIF, respectively.
> >
> > In our framework, the primary focus is on the final output rather than intermediate steps, as our goal is not to synthesize materials but to generate accurate final representations. Here are some key points regarding our approach:
> >
> > **Lack of Intermediate Labels:** We currently do not have accurate true labels for intermediate steps, which makes it challenging to evaluate success rates at the intermediate stage individually.
> >
> > **Utilizing LLM for Bridging Steps:** We use GPT-4 to bridge these intermediate steps for training, which helps generate coherent pathways and reasoning that connect input queries to final outputs effectively in practice.
> >
> > **Interpretable Reasoning with LLM:** By employing LLMs, we incorporate structured and interpretable reasoning into the generation process. This not only enhances the accuracy and reliability of the outputs but also provides insights into how each decision is made, making the process more transparent.
> >
> > In the future, we plan to consider both intermediate reasoning and results more comprehensively. This will allow us to evaluate and improve each step of the process, potentially leading to more refined and accurate outputs.

---

> > > ### Author Response · Authors · 2024-11-20
> > >
> > > > **Q3** Usefulness for User Queries
> > >
> > > **AQ3** One of the key advantages of our LLM-based material generation approach over traditional approaches like CDVAE is its ability to handle open queries with diverse conditions. Our retrieval model is trained with a tool named Robocrystallographer, which encompasses a comprehensive range of properties extracted from materials in the database. As a result, our model is not dependent on specific prompts or properties/conditions. During the training stage, we randomly select properties to construct our prompts, allowing for flexibility and adaptability. We only fix the set of conditions during the evaluation stage to assess specific properties.
> > >
> > > Additionally, the base Llama model includes foundational knowledge about chemistry and materials, which may help infer properties not explicitly present in our training database. This inherent knowledge enables the model to make educated predictions about unfamiliar properties, further enhancing its capability to respond to a wide array of user queries. This flexibility ensures that users can query the model with broad or specific criteria, greatly enhancing its utility in material discovery.
> > >
> > > > **Q4** Response to Non-Existent Materials
> > >
> > > **AQ4** If a target material does not exist, MatExpert suggests the closest existing materials with potential modifications to approach the desired properties and may produce unsuccessful examples. By performing multiple inference rounds, the model can explore a broader range of possibilities. This is a natural challenge for all baselines and state-of-the-art methods in material generation.
> > >
> > > For future work, we can enhance the model by outputting confidence probabilities for the generated examples. This will allow users to evaluate the likelihood of correctness for each generated material. Users can then assess the generated materials based on both the Chain-of-Thought (CoT) reasoning/pathway and the associated confidence levels, providing a more informed and reliable basis for decision-making.

---

> > > > ### Comment · Reviewer_SCz7 · 2024-11-30
> > > >
> > > > Thank you for your response. I appreciate the clarification and the updated manuscript. However, I am still not sure about the design choice between RAG retrieval of chemical structure vs natural language. If the authors can provide a stronger justification for why chemical structures offer a significant advantage over NLP corpora, it would greatly enhance the perceived novelty. For instance, do chemical structures provide a more compact representation without losing the information conveyed in textual descriptions?
> > > >
> > > > As for  chain-of-thought, its novelty in this context appears somewhat incremental, particularly given the demonstrated improvements of CoT in prior works. To better highlight the distinctiveness of your approach, it would be helpful to elaborate on how the CoT reasoning applied here diverges from or expands upon existing methods.

---

> > > > > ### Author Response · Authors · 2024-12-01
> > > > >
> > > > > We greatly appreciate the reviewer’s insightful feedback. We would like to take this opportunity to clarify the comparison to HoneyComb and LLaMP (both are natural language-based RAG), the novelty of MatExpert (structure retrieval). Then we clarify the novelty of our CoT approach.
> > > > >
> > > > > **1. Comparison to HoneyComb and LLaMP**
> > > > >
> > > > > We would like to clarify the different utilizations of Retrieval-Augmented Generation (RAG) in our work compared to HoneyComb and LLaMP:
> > > > >
> > > > > * **HoneyComb**: RAG in HoneyComb is primarily used for retrieval and tool-based analysis of existing materials. While it enhances reasoning and allows access to external tools for detailed computations, the focus remains on understanding and analyzing existing materials, not on generating new ones. RAG is used here to retrieve material knowledge and perform simulations, rather than to create new materials.
> > > > >
> > > > > * **LLaMP**: LLaMP uses RAG to retrieve high-fidelity material data and then simulate or predict material properties based on this data. It employs hierarchical agents for reasoning and improving predictions, but again, the core task remains analysis and simulation, not material generation. The RAG framework in LLaMP helps improve the accuracy of predictions rather than enabling the creation of new materials.
> > > > >
> > > > > * **MatExpert**: We employ structure retrieval to generate novel material structures. In the retrieval stage, we identify a relevant base material, which is then refined in the transition phase to meet the user’s specifications. This iterative process of generation and modification directly leads to the design of new materials based on specific user-defined properties. Structure retrieval is critical in facilitating this transition from existing knowledge to novel material creation.
> > > > >
> > > > > **2. Why We Focus on Structure Retrieval:**
> > > > >
> > > > > Our work emphasizes structure retrieval rather than RAG for several key reasons:
> > > > >
> > > > > * **Differences in tasks:** The key task difference between MatExpert and methods like HoneyComb and LLaMP is that MatExpert focuses on generating new materials. In contrast, HoneyComb and LLaMP focus on retrieving and analyzing data from existing documents. HoneyComb uses its tools and knowledge base to answer queries and perform simulations on existing materials, while LLaMP enhances the process of retrieving and predicting properties from large datasets. MatExpert’s task is not merely to analyze or simulate existing materials but to create new materials that meet specific properties defined by the user. This generative task requires a more direct and structure-based approach to retrieving materials with specific property-structure relationships, which is why structure retrieval is more suitable than RAG in this context.
> > > > >
> > > > > * **Data Characteristics:** Structured materials databases (e.g., the NOMAD or Materials Project) offer a reliable, consistent, and well-organized source of property-structure information. This makes structure retrieval a more direct and effective approach for our specific task of material generation rather than traditional RAG.
> > > > >
> > > > > We want to emphasize that our current focus on structure retrieval does not undermine the value of RAG. Instead, we view RAG and structure retrieval as complementary approaches, and MatExpert’s future work will seek to integrate both:
> > > > >
> > > > > * **Integrating Structure Retrieval with RAG:** Although we currently prioritize structure retrieval for material generation, we plan to integrate language-based RAG to further augment the retrieval process. This integration will expand the retrieval space, allowing us to more effectively combine textual and structural information, enhancing the overall material discovery process.
> > > > >
> > > > > * **Enhancing Reasoning with RAG:** RAG, leveraging vast textual corpora (such as Arxiv papers, Wikipedia, Python REPL, etc.), can provide valuable explanatory insights in the form of property-structure relationships, causal links, and correlations. These can enhance our reasoning process during the transition stage, enabling us to generate more robust and interpretable pathways for creating new materials.

---

> ### Author Response · Authors · 2024-12-01
>
> **3. What’s the Novelty of Our CoT**
>
> We acknowledge that CoT reasoning has been widely studied and demonstrated to improve reasoning tasks in prior works. However, our method introduces distinct advancements tailored to the material generation context, which we elaborate on below:
>
> **1) Traditional CoT with Prompts**:
>
> * In traditional CoT methods, CoT reasoning is often achieved through prompt engineering, where step-by-step reasoning is embedded directly in the input prompts. These prompts guide the LLM to simulate a logical reasoning process for problem-solving tasks, such as multi-step mathematical or scientific reasoning.
>
> * While effective, these CoT prompts typically rely on implicit reasoning paths derived from a single query-response interaction. They do not benefit from external, validated reference steps or explicit intermediate feedback loops.
>
> **2) Our CoT with GPT-4-Generated Ground-Truth Pathways**:
>
> In contrast, our approach leverages GPT-4 to generate detailed, step-by-step ground-truth transition pathways during the training phase. These pathways are not limited to a single static prompt but are derived from:
>
> * A carefully designed training dataset of <source material, target material, transition pathway> triples.
>
> * GPT-4’s generation capabilities, which provide a comprehensive breakdown of how the material properties and structures should be modified to meet the target specifications.
>
> By using these ground-truth pathways, we explicitly integrate validated intermediate steps into the CoT reasoning process, enabling our system to replicate expert-like reasoning for the iterative modification of materials.
>
> We hope this clarification addresses your concern regarding the novelty of MatExpert. Please let us know if this response resolves your questions or if further clarification is needed.

---

> > ### Comment · Reviewer_SCz7 · 2024-12-02
> >
> > Thank you for the detailed clarification on the comparison of different methods. However, I am still not conviced by the design choice of structure retrieval. Could the authors provide additional details on how chemical structures offer more compact or meaningful information than textual data for the material generation task studied in this paper? To my understanding, textual data should inherently provide richer contextual information than chemical structures alone.

---

> > > ### Author Response · Authors · 2024-12-03
> > >
> > > Thank you for your thoughtful feedback and for raising this question about the design choice of structure retrieval. We provide additional clarification on the advantages of leveraging chemical structures and the rationale behind our design.
> > >
> > > > **1. Our Pipeline: Retrieved Structure → Textual Description → Prompt for LLMs**
> > >
> > > Our framework does not rely directly on chemical structures alone. Instead, we integrate structure retrieval with natural language conversion to enhance the contextual richness of the information used in material generation.
> > >
> > > Specifically, the retrieved structures are processed using the tool robocrystallographer, which converts structural information into natural language descriptions. These descriptions include contextual details such as space group, atomic arrangements, and bonding environments.
> > > The resulting textual descriptions, combined with the property data, are incorporated into the input prompts in the second stage of our workflow to guide the material generation process.
> > > For reference, see Lines 239–244 in the manuscript, where `<description_source>` represents the natural language description extracted by robocrystallographer.
> > >
> > > This pipeline ensures that the structured data is converted into rich textual descriptions, combining the precision of structured information with the contextual richness required for effective input to large language models (LLMs).
> > >
> > > > **2. The Challenge of Learning Property-Structure Patterns**
> > >
> > > In material generation, a critical task involves identifying and leveraging implicit patterns between properties and structures. These patterns—such as how specific atomic configurations affect stability, bandgap, or other material properties—are often intricate and challenging to discern. Our framework addresses this challenge by employing supervised contrastive learning, which offers distinct advantages over RAG-based methods:
> > >
> > > **1) Advantages of Supervised Contrastive Learning**
> > >
> > > * **Aligned Representations:** Supervised contrastive learning aligns property and structure representations by learning from labeled pairs of property descriptions and structural data.
> > > Embedding Consistency: Materials with similar properties are mapped closer together in the embedding space, while dissimilar materials are mapped farther apart. This alignment fosters a robust understanding of property-structure relationships, enabling effective generalization for material generation tasks.
> > >
> > > * **Structured Data Benefits:** Using structured, noise-free data from material databases ensures consistency during training, making it easier to capture and model these intricate relationships reliably.
> > >
> > > **2) Limitations of training-free RAG for Implicit Pattern Learning**
> > >
> > > * **Lack of Explicit Alignment:** RAG-based methods rely on textual corpora, which often lack explicitly aligned property-structure relationships. Textual data is frequently noisy, fragmented, or contextually ambiguous, making it challenging to infer how specific material properties correlate with atomic structures.
> > >
> > > * **Training-Free Nature:** Unlike supervised contrastive learning, RAG is inherently training-free, relying on pre-trained language models and external tools. While useful for general retrieval tasks, this approach is not optimized for learning and generalizing the precise property-structure mappings required for material generation.
> > >
> > > > **3. Case Study**
> > >
> > > To illustrate the advantages of our approach, we present a case study comparing the outputs of MatExpert and the state-of-the-art RAG-based method HoneyComb.
> > >
> > > We provide the outputs from structure retrieval (used in MatExpert) and RAG-based retrieval (HoneyComb) for the same query. This comparison highlights how structure retrieval delivers precise and property-aligned outputs, while RAG-based retrieval may struggle to achieve similar specificity.
> > >
> > > (next page)

---

> ### Author Response · Authors · 2024-12-03
>
> > **Inputs for MatExpert and HoneyComb**
>
> ```
> Below is a description a bulk material: The formation energy per atom is -1.828. The band gap is 1.1081. The energy above the convex hull is 0.0491. The elements are Co, Li, Mn, Ni, O. The spacegroup number is 5.
>
> Based on the information, could you generate a description of the lengths and angles of the lattice vectors and then the element type and coordinates for each atom within the lattice?
> ```
>
> **1. HoneyComb**
>
> > **RAG outputs from HoneyComb**
>
> ```
> ['"The answer to the question by running pre-defined tools, is: The information extracted from the academic papers will be used to generate a description of the lengths and angles of the lattice vectors and the element type and coordinates for each atom within the lattice.. The following are the tools used and the responses: [(AgentAction(tool=\'Arxiv Search\', tool_input=\'crystal structure lattice vectors elements coordinates\', log=\'This question requires understanding of crystallography and material properties. I should search for academic papers or articles related to crystal structures and lattice vectors.\\\\nAction: Arxiv Search\\\\nAction Input: \\"crystal structure lattice vectors elements coordinates\\"\'), \'Published: 2015-03-25\\\\nTitle: Crystal Structure Representations for Machine Learning Models of Formation Energies\\\\nAuthors: Felix Faber, Alexander Lindmaa, O. Anatole von Lilienfeld, Rickard Armiento\\\\nSummary: We introduce and evaluate a set of feature vector representations of crystal\\\\nstructures for machine learning (ML) models of formation energies of solids.
> ...
> Here, is the fraction of the grain boundary monolayer available for segregated atoms at saturation, is the actual fraction covered with segregant, is the bulk solute molar fraction, and is the free energy of segregation per mole of solute. Values of were estimated by McLean using the elastic strain energy , , released by the segregation of solute atoms. The solute atom is represented by an elastic sphere fitted into a spherical hole in an elastic matrix continuum. The elastic energy associated with the solute atom is given by: where is the solute bulk modulus , is the matrix shear modulus , and and are the atomic radii of the matrix and impurity atoms, respectively. This method gives values correct to within a factor of two (as compared with experimental data for grain boundary segregation), but a greater accuracy is obtained using the method of Seah and Hondros, described in the following section. Free energy of grain boundary segregation in binary systems Using truncated BET theory (the gas adsorption theory developed by Brunauer, Emmett, and Teller), Seah and Hondros write the solid-state analogue as: where is the solid solubility , which is known for many elements (and can be found in metallurgical handbooks). In the dilute limit, a slightly soluble substance has , so the above equation reduces to that found with the Langmuir-McLean theory. This equation is only valid for . If there is an excess of solute such that a second ph\\""']
> ```
>
> > **Final outputs from HoneyComb**
>
> ```
> To generate a description of the lengths and angles of the lattice vectors and the element type and coordinates for each atom within the lattice, specific crystallographic data is required. This includes detailed information about the unit cell dimensions, angles between lattice vectors, and atomic positions within the unit cell. The provided information about the bulk material, such as formation energy, band gap, energy above the convex hull, elements involved, and spacegroup number, does not include these specific crystallographic details.
>
> 1. **Lattice Vectors and Angles**: The spacegroup number (5) indicates a specific symmetry, but without additional data such as unit cell parameters (a, b, c) and angles (, , ), the exact lengths and angles of the lattice vectors cannot be determined.
>
> 2. **Atomic Positions**: Similarly, while the elements present (Co, Li, Mn, Ni, O) are known, the exact coordinates of these atoms within the lattice require detailed structural data, which is not provided.
>
> In conclusion, the necessary information to describe the lattice vectors and atomic coordinates is not available in the provided text. Therefore, it is not possible to generate a description of the lattice structure based solely on the given information.
> ```
>
> **2. MatExpert**
>
> (Please refer to Figure 3 in our manuscript to view the results of the structure retrieval and the final outputs.)
>
> This comparison highlights how structure retrieval delivers precise and property-aligned outputs, whereas RAG-based retrieval may struggle to achieve the same level of specificity and may fail to adequately respond to the query.
>
> We hope this clarification addresses your concern regarding the design choice of structure retrieval. Please let us know if this response resolves your questions or if further clarification is needed.

---

> > ### Comment · Reviewer_SCz7 · 2024-12-03
> >
> > Thank you for the detailed clarification and case study. These results strengthen the work and have clarified my concerns. I appreciate the effort into improving the work and I updated my score accordingly. Thank you!

---

> > > ### Author Response · Authors · 2024-12-03
> > >
> > > Thank you for taking the time to carefully review our responses and for raising your score. We sincerely appreciate your thoughtful feedback on the novelty and design choices in our work, the elaboration and clarification of figures, as well as your recognition of our efforts to address your concerns.
> > >
> > > We are committed to continuing to improve our research and are encouraged by your thoughtful insights. Thank you once again for your consideration. Your insights have been invaluable in helping us refine and strengthen our research, and we are truly grateful for your support.

---

### Official Review · Reviewer_vZKP · 2024-11-04

**Soundness:** 2
**Presentation:** 3
**Contribution:** 3
**Rating:** 6
**Confidence:** 3

**Summary:**

The paper describes a novel LLM based framework for materials discovery (MatExpert). There are three stages to the MatExpert methodology: retrieval, transition, and generation. First, a material is retrieved from the database that most closely matches the description given (similarly was trained with contrastive learning). Next during transition, the model determines how to alter the retrieved material to match the desired properties. Lastly, in the generation phase the model produces a ALX representation that is converted into a CIF representation. The main contribution of the paper is the MatExpert framework and the accompanying benchmarking and ablation study.

**Strengths:**

- The application of LLM to materials is interesting and materials discovery is important
- The evaluation metrics includes stability computed with DFT not just proxy metrics
- Writing style and related work are good

**Weaknesses:**

- Lacking details on what data was used for which tasks? For the unconditional results on MP-20, it is unclear if the NOMAD data was also used for training MatExpert. For the conditional results, were CrystalLLM and MatExpert trained on the same data?
- The results in Figure 5 are not well quantified i.e. it is not clear MatExpert is better. Also, there are 11 bars but only 9 labels, not sure if I missed something? The colors are very similar in some cases, hard to parse quickly.
- There is no discussion on the limitations of using the retrieval stage. My interpretation is that the retrieved material is like a template. Before generative models, new materials were searched for using templating/substitution methods. One of the critiques of those methods is that materials generated are still quite similar to the template, would that also be a limitation here?

**Questions:**

- The paper mentions that the wdist density and wdist number of elements metrics are greatly improved compared to CrystalLLM but if the model is given a template from the database does that undermine these metrics? Is there a way to test this? For example, how often does the generated structure change the number of elements compared to the retrieved/given material?
- Can you compute the S.U.N metric from Zeni et al. (https://arxiv.org/abs/2312.03687)?
- How does the inference speed of MatExpert compare to CrystalLLM?

---

> ### Author Response · Authors · 2024-11-20
>
> We appreciate the valuable feedback and have carefully considered the points raised. Below, we address your concerns. We hope our revisions clarify all the questions and strengthen the quality of our work.
>
> > **W1** Data Usage Clarification
>
> **AW1** We apologize for any confusion regarding the data used for training and testing. To clarify:
>
> 1. For unconditional generation, MatExpert was trained exclusively on the MP-20 dataset, aligning with the datasets used by other baseline models.
>
> 2. For conditional generation, MatExpert was trained exclusively on the NOMAD dataset.
>
> It is important to note that MatExpert was not trained on both datasets simultaneously. We ensure that our comparisons remain fair and consistent across different models and tasks. We originally stated this in lines 315–317 and lines 389–392. The clarification is now further included in line 316, line 321, and line 370 in the updated manuscript.
>
> > **W2** Figure 5 Clarification
>
> **AW2** Thank you for pointing out the issues with Figure 5. The last two bars represent MatExpert-80B ($\tau$=1.0) and MatExpert-80B ($\tau$=0.7). We also added numerical values on the top of each bar for clarity and included missing legends. Please check the figure in the updated manuscript.
>
> From the figure, we can observe that:
> + MatExpert consistently achieves high scores in both structure and composition diversity compared to CDVAE and various Crystal-LLM configurations, indicating its superior ability to explore a broad chemical space.
> + MatExpert demonstrates the capacity to generate novel structures and compositions aligned with the distribution of the test sets, as shown by its consistent novelty scores close to the test set.
> + Crystal-LLM faces challenges with larger models exhibiting lower novelty. In contrast, MatExpert consistently maintains high novelty across different model sizes.
>
> > **W3** Retrieval Stage Limitations
>
> **AW3** Our method uses the retrieved material as a template, which inherently supports both the benefits and challenges of this strategy:
>
> **Role of the Template:** The retrieved material serves as a foundational template, allowing the model to leverage existing structures that align closely with desired properties. This can enhance the accuracy of generated materials by providing a realistic starting point.
>
> **Trade-offs and Limitations:** There is a trade-off between the extent of modifications possible and the quality of the predicted properties. While larger modifications may be more challenging and could impact accuracy, small modifications often maintain desirable characteristics of the original material and allow for incremental improvements.
>
> **Observations and Empirical Evidence:** Empirically, we have observed that the limitations of this approach are not significant in our current framework. The use of templates generally supports successful material generation without substantial drawbacks.
>
> **Avoiding Cumulative Errors:** Since the process of MatExpert involves only two rounds of conversation with one step of refinement, minimizing the potential for error accumulation. Our empirical results show that despite taking the risk of error propagation, the overall performance improvements justify the multi-stage approach.
>
> **Future Improvement:** With additional computational resources, we can have the following improvement directions: 1) implement feedback loops to allow for iterative refinements, thus addressing any initial inaccuracies in the retrieval stage. 2) With tools like pymatgen and m3gnet, we can check whether the retrieved material at the initial step is similar to the requested property constraints to avoid error propagation. Additionally, we only use a very trivial retrieval approach with only one step of refinement to demonstrate the effectiveness of the multi-stage process, we believe with the most advanced RAG technique, it will be further improved.

---

> > ### Author Response · Authors · 2024-11-20
> >
> > > **Q1** wdist Metrics and Template Influence
> >
> > **AQ1** The wdist metrics measure the Wasserstein distance in terms of density and number of elements, providing insights into the distribution of generated materials relative to the test set. The templates used in our retrieval stage are sourced exclusively from the training set. We then evaluate the generated materials by calculating the distance between them and the test set, ensuring there is no overlap between the templates and the test set. Since the training and test sets are randomly sampled from the entire dataset, we acknowledge that the use of a template in the retrieval stage may contribute to MatExpert achieving lower wdist density and wdist number of elements metrics compared to CrystalLLM. This highlights the superiority of the retrieval stage in MatExpert, as it effectively guides the generation process toward more accurate material distribution.
> >
> > > **Q2** S.U.N Metric
> >
> > **AQ2** Thank you for suggesting the computation of the S.U.N metric from Zeni et al. The S.U.N metric, which stands for Stability, Uniqueness, and Novelty, is not currently publicly available with any open-source codes. However, we address similar aspects in our evaluation by including stability and novelty metrics, as detailed in our manuscript. We have calculated the S.U.N percentage for our generated materials by the definition of S.U.N to be:
> >
> > | Methods                 | Percentage of S.U.N |
> > |-----|--------|
> > | MatExpert 7B   (\tau=0.7) | 3.9 |
> > | MatExpert 7B (\tau=1.0)   | 3.9 |
> > | MatExpert 70B (\tau=0.7)  | 4.2 |
> > | MatExpert 70B (\tau=1.0)  | 4.4 |
> >
> >
> > > **Q3** Inference Speed Comparison
> >
> > **AQ3** Here’s the detailed inference speed comparison:
> >
> > **Comparative Complexity:** We have two important baselines: CDVAE and CrystaLLM. Compared to diffusion-based methods like CDVAE, our approach is more efficient than their complex diffusion processes. Compared to one-step LLM CrystaLLM, our method does have higher computational complexity, but as we only have 2 rounds with around 2 times more time costs in inference. Below is a summary of the inference speed for these methods:
> >
> > | Methods                 | Inference Speed |
> > |-----|--------|
> > | CDVAE                 | ~2.1 sec / sample |
> > | CrystaLLM 70B         | ~1.4 sec / sample |
> > | MatExpert 70B (ours)  | ~1.8 sec / sample |
> >
> > Please note that due to differences in hardware, inference library implementations, and hyperparameters such as the number of iterative steps in CDVAE, the computational costs are provided only for reference.
> >
> > **Better performance:** As our framework operates offline rather than in real-time, it is important to consider the trade-off between efficiency and performance. While our method involves a multi-stage process, it does not incur exponentially higher time costs compared to other state-of-the-art methods. Instead, it achieves significantly better performance, demonstrating the superiority of our approach in balancing computational efficiency with enhanced material generation quality.
> >
> > **Efficient Measures and Further Improvement:** We have implemented efficient algorithms, such as FlashAttention for the 70B model, to streamline the process and reduce computational overhead without compromising performance. We are open to further improving computational efficiency, especially as more open-source resources like efficient LLM inference libraries become available.

---

> > > ### Comment · Reviewer_vZKP · 2024-11-27
> > >
> > > Thanks for clarifying what training data was used. Also, the update to Fig 5 is nice. There are clear advantages to using a template from the retrieval stage in terms of generating reasonable structures, however it is also desirable to find materials with very different structures that may have the properties of interest i.e. there are inherent limitations to using templating methods, it would be nice to have some discussion of this in the paper. Overall the empirical results show an improvement over existing methods, so I lean towards acceptance. I have updated my score accordingly.

---

> > > > ### Author Response · Authors · 2024-11-27
> > > >
> > > > Thank you for your thoughtful review and for taking the time to consider our responses. We appreciate your feedback and are glad that the clarifications regarding data usage and Figure 5 were helpful.
> > > >
> > > > We acknowledge your point about the limitations inherent in using templating methods for materials discovery. As you suggested, we will incorporate a more detailed discussion of this aspect in the paper to provide a balanced view of the strengths and potential constraints of the retrieval stage. This will help readers better understand the trade-offs involved in our approach.
> > > >
> > > > Looking forward, we plan to explore strategies to mitigate the limitations of templating methods, such as implementing advanced retrieval-augmented generation (RAG) techniques that can potentially enhance the diversity of generated materials. Additionally, we are considering iterative refinement processes that could help discover materials with significantly different structures while maintaining desired properties.
> > > >
> > > > Thank you again for your constructive feedback and your consideration towards the acceptance of our work. Your insights are invaluable in helping us improve our manuscript.

---

### Official Review · Reviewer_Krbm · 2024-11-08

**Soundness:** 3
**Presentation:** 3
**Contribution:** 2
**Rating:** 6
**Confidence:** 3

**Summary:**

MatExpert is designed to streamline the discovery of new materials using LLMs and contrastive learning. Inspired by the traditional workflow of human experts, MatExpert operates in three stages: retrieval, transition, and generation. Experimental results demonstrate that MatExpert outperforms current sota models in material generation tasks.

**Strengths:**

1. The design of MatExpert mirrors the expert-driven process in material science, breaking down material generation into retrieval, transition, and generation stages. This structured approach allows for iterative refinement.

2. The transition stage uses a CoT reasoning process, enabling the model to outline logical, step-by-step modifications to meet target properties. This sequential reasoning contributes to the model's ability to achieve high accuracy in conditional generation tasks.

3. By compiling a dataset of over 2 million materials from NOMAD, MatExpert provides a large-scale testbed to assess its performance.

**Weaknesses:**

1. While the multi-stage design of MatExpert improves accuracy, it adds computational complexity and potentially increases training time compared to single-step models.

2. The proposed framework will have cumulative errors. If the result retrieved in the first step is far away from the target, it will be difficult to correct it later, thus affecting the results of subsequent steps.

3. This paper focuses on innovation in application scenarios, and the technological innovation is relatively limited.

**Questions:**

The heavy reliance on specific material databases, such as NOMAD, might lead to overfitting or model bias toward these datasets. Testing MatExpert on unseen data sources or a wider array of material properties could offer better insights into its generalization capabilities.

---

> ### Author Response · Authors · 2024-11-20
>
> We appreciate the valuable feedback and have carefully considered the points raised. Below, we address your concerns. We hope our revisions clarify all the questions and strengthen the quality of our work.
>
> > **W1** Concern about computational complexity
>
> **AW1** We acknowledge that the multi-stage design may increase computational complexity compared to single-step models. Here’s how we consider this issue:
>
> **Comparative Complexity:** We have two important baselines to consider: CDVAE and CrystaLLM. Our approach is more efficient than diffusion-based methods like CDVAE, as it avoids the complexity inherent in diffusion processes. While our method does have higher computational complexity than the one-step LLM approach used by CrystaLLM, it only requires two rounds of inference, resulting in approximately twice the time cost. Below is a summary of the inference speed for these methods:
>
> | Methods                 | Inference Speed |
> |-----|--------|
> | CDVAE                 | ~2.1 sec / sample |
> | CrystaLLM 70B         | ~1.4 sec / sample |
> | MatExpert 70B (ours)  | ~1.8 sec / sample |
>
> Please note that due to differences in hardware, inference library implementations, and hyperparameters such as the number of iterative steps in CDVAE, the computational costs are provided only for reference.
>
> **Better performance:** As our framework operates offline rather than in real-time, it is important to consider the trade-off between efficiency and performance. While our method involves a multi-stage process, it does not incur exponentially higher time costs compared to other state-of-the-art methods. Instead, it achieves significantly better performance, demonstrating the superiority of our approach in balancing computational efficiency with enhanced material generation quality.
>
> **Efficient Measures and Further Improvement:** We have implemented efficient algorithms, such as FlashAttention for the 70B model, to streamline the process and reduce computational overhead without compromising performance. We are open to further improving computational efficiency, especially as more open-source resources like efficient LLM inference libraries become available.
>
> > **W2** Concern about cumulative errors of multi-stage process.
>
> **AW2** We acknowledge the concern regarding cumulative errors in the multi-stage process. Here’s how we consider this issue:
>
> **Only limited rounds:** Since the process of MatExpert involves only two rounds of conversation with one step of refinement, the potential for error accumulation is minimized. Our empirical results show that despite taking the risk of error propagation, the overall performance improvements justify the multi-stage approach.
>
> **Further Improvement:** With additional computational resources, we can have the following improvement directions: 1) implement feedback loops to allow for iterative refinements, thus addressing any initial inaccuracies in the retrieval stage. 2) With tools like pymatgen and m3gnet, we can check whether the retrieved material at the initial step is similar to the requested property constraints to avoid error propagation. Additionally, we only use a very trivial retrieval approach with only one step of refinement to demonstrate the effectiveness of multi-stage process, we believe with the most advanced RAG technique, it will be further improved.

---

> > ### Author Response · Authors · 2024-11-20
> >
> > > **W3** Concern about technical innovation
> >
> > **AW3** While the primary focus of our work is on application innovation, we believe there are several points of technological innovation:
> >
> > **Novel Paradigm:** Our framework introduces a new paradigm for material discovery by integrating retrieval, refinement, and generation stages. Specifically in the context of conditional material generation, MatExpert demonstrates remarkable flexibility and ability to generate novel material conditioning on the user-predefined constraints on properties with this new paradigm and finetuned LLMs. To the best of our knowledge, it is the first to work with this paradigm and demonstrate effectiveness on large-scale material benchmark datasets.
> >
> > **Novel Retrieval Approach:** Our retrieval approach, the first step of MatExpert, is different from traditional retrieval-augmented generation (RAG) approaches commonly used in natural language processing (NLP). Traditional RAG methods typically focus on retrieving relevant text snippets or knowledge pieces from large NLP corpora to enhance the generation process. In contrast, our approach is tailored specifically for material generation, where we retrieve material structures from databases instead of textual information. This structural retrieval serves as a seed for subsequent modifications, providing a tangible and scientifically grounded starting point for material generation.
> >
> > **Novel Cain-of-Thought Approach:** In our framework, we use a Chain-of-Thought (CoT) reasoning approach to enhance the transition stage. GPT-4 takes both the true material structure (answer) and the user query as inputs to generate detailed modification steps (thought). These steps serve as ground-truth pathways for fine-tuning LLaMA. LLaMA, in turn, takes the query alone as input and is trained to generate both the modification pathways (thought) and the resulting material structure (answer). This approach enables LLaMA to apply expert-like reasoning, ensuring accurate and effective material modifications. With the inclusion of contrastive learning and CoT reasoning, our framework provides interpretable pathways for material modifications, allowing users to evaluate the reliability and reasonableness of generated outcomes.
> >
> > **Dataset Innovation:** Our work collects and leverages the NOMAD database, comprising over 2.8 million materials, to set a new standard in material generation models. This extensive dataset, larger and more diverse than those typically used, like MP-20 or Carbon-24, reduces the risk of overfitting and enhances model generalization. Our approach not only improves material generation performance but also encourages the use of large-scale, high-quality datasets for more robust discoveries in materials science.
> >
> > > **Q1** Model Generalization and Dataset Reliance
> >
> > **AQ1** As the response **AW3**, we emphasize that our work utilizes the NOMAD database, one of the largest and most diverse materials datasets available, to ensure a comprehensive training base that enhances model generalization. To maintain a fair comparison with other models, we clarify our training datasets as follows:
> >
> > 1. For unconditional generation, MatExpert was trained exclusively on the MP-20 dataset, aligning with the datasets used by other baseline models.
> >
> > 2. For conditional generation, MatExpert was trained exclusively on the NOMAD dataset.
> >
> > It is important to note that MatExpert was not trained on both datasets simultaneously. This approach ensures that our comparisons remain fair and consistent across different models and tasks.
> >
> > By being the first to train on such a large-scale dataset like NOMAD, we are setting a new benchmark for evaluating model performance in materials science, and we encourage future research to adopt similarly comprehensive datasets to drive further advancements in the field. In the future, we will test MatExpert on additional datasets and incorporate a wider range of material properties to better evaluate its generalization capabilities.

---

### Meta-Review · Area_Chair_77Ww · 2024-12-17

**Metareview:**

The paper introduces MatExpert, a framework that leverages Large Language Models (LLMs) and contrastive learning to streamline the discovery of new solid-state materials. By emulating expert workflows through retrieval, transition, and generation stages, MatExpert demonstrates superior performance over existing methods on large-scale datasets like NOMAD.

**[Strengths]**
- The three-stage approach effectively mimics expert-driven material discovery, enhancing generation quality.
- MatExpert outperforms state-of-the-art models in validity, distribution alignment, and stability metrics.

**[Weaknesses]**
- Sequential stages may lead to cumulative errors if the initial retrieval is suboptimal.
- The multi-stage process results in higher computational overhead compared to single-step models.

**[Decision]**
The authors extensively addressed most of the reviewers’ concerns in their rebuttal. I recommend accepting the paper. I encourage the authors to integrate the addressed concerns into the final manuscript.

**Additional Comments On Reviewer Discussion:**

During the rebuttal period, reviewers raised concerns about the computational complexity and potential error propagation inherent in the multi-stage design of MatExpert. Questions were also directed towards the novelty of using structure retrieval over traditional RAG methods and the limited technological innovations presented. Additionally, the dependence on the NOMAD dataset was scrutinized for potential biases.

 The authors addressed these points by providing detailed computational comparisons, clarifying data usage and training protocols, and justifying their methodological choices with additional experiments and updated figures. They also acknowledged the limitations and proposed future improvements, which satisfactorily alleviated the reviewers’ concerns. Consequently, the collective positive responses and effective rebuttal led to the decision to accept the paper.

---

### Decision · Program_Chairs · 2025-01-22

Accept (Poster)